# Loss-of-function mutations in the dystonia gene THAP1 impair proteasome function by inhibiting PSMB5 expression

Dylan E. Ramage, Drew W. Grant & Richard T. Timms ✉

The 26S proteasome is a multi-catalytic protease that serves as the endpoint for protein degradation via the ubiquitin-proteasome system. Proteasome function requires the concerted activity of 33 distinct gene products, but how the expression of proteasome subunits is regulated in mammalian cells remains poorly understood. Leveraging coessentiality data from the DepMap project, here we characterize an essential role for the dystonia gene *THAP1* in maintaining the basal expression of *PSMB5*. PSMB5 insufficiency resulting from loss of THAP1 leads to defects in proteasome assembly, impaired proteostasis and cell death. Exploiting the fact that the toxicity associated with loss of THAP1 can be rescued upon exogenous expression of PSMB5, we define the transcriptional targets of THAP1 through RNA-seq analysis and perform a deep mutational scan to systematically assess the function of thousands of single amino acid THAP1 variants. Altogether, these data identify THAP1 as a critical regulator of proteasome function and suggest that aberrant proteostasis may contribute to the pathogenesis of THAP1 dystonia.

Regulated protein degradation is essential for cellular homeostasis. As the primary route through which the cell achieves selective protein degradation, the ubiquitin-proteasome system (UPS) plays an important role in essentially all critical cellular processes[1]. Proteins destined for degradation are typically identified by E3 ubiquitin ligases, which, following activation of ubiquitin by an E1 activating enzyme and its transfer to an E2 ubiquitin conjugating enzyme, catalyze the attachment of ubiquitin onto substrate proteins[2]. Addition of further ubiquitin moieties generates polyubiquitin chains, which can serve as a potent recognition signal for the 26S proteasome, a multi-catalytic protease. The importance of this pathway is underscored by the fact that dysregulation of the UPS is a hallmark of diseases such as cancer, autoimmunity and neurodegeneration[3].

The 26S proteasome is a large, multi-subunit complex comprising the 20S core particle and two 19S regulatory particles[4]. The regulatory subunits are responsible for the recognition and unfolding of ubiquitinated proteins, which are then threaded into the active site in the core particle formed by two rings of β-subunits[5]. The complex comprises three catalytic subunits: PSMB5 (also known as β5), PSMB6 (β1)

and PSMB7 (β2). These exhibit trypsin-like, chymotrypsin-like and caspase-like activities respectively, resulting in the proteolysis of the polypeptide chain into short peptides fragments[6]. Each catalytic subunit harbors a catalytic threonine residue at its N-terminus[6], which is activated following autocatalytic processing of an N-terminal propeptide at a late stage in core particle assembly[7].

Transcriptional regulation of proteasome subunit expression is known to be controlled by two major players. In yeast the transcription factor Rpn4 is responsible for both the basal expression of proteasome subunits, and, under conditions of proteasome insufficiency, feedback induction of proteasome subunit expression[8]. Rpn4 functions as part of a negative feedback loop that monitors proteasome activity: in unstressed cells Rpn4 is constitutively degraded, but it is rapidly stabilized upon proteasome dysfunction[9]. In mammalian cells Nrf1 acts in a similar manner to induce the expression of proteasome subunits under conditions of proteasome insufficiency[10,11], but appears not to have a major role in their basal expression[10]. Several other transcription factors have been implicated in proteasome gene expression[12–14], including NF-Y which regulates a set of proteasome genes which carry

Cambridge Institute of Therapeutic Immunology and Infectious Disease, Department of Medicine, University of Cambridge, Puddicombe Way, Cambridge, UK. ✉e-mail: rtt20@cam.ac.uk

a CCAAT box motif in their promoters[14], but the factors which maintain the basal expression of proteasome subunits in human cells remain largely unknown.

By performing genome-wide pooled CRISPR/Cas9 loss-of-function genetic screens across hundreds of cancer cell lines, the Broad Institute's Cancer Dependency Map (DepMap) project aims to systematically catalog the essentiality of all protein-coding human genes[15]. A key insight from these data is that whilst the dependency of different cell lines on any one particular gene may vary, genes which function in concert in a biological pathway often exhibit globally similar essentiality patterns[16]. Thus, by measuring gene dependency across hundreds of cell lines, genes exhibiting 'co-essential' relationships can be clustered into modules which may have the power to predict novel functions for genes. Indeed, multiple studies have exploited this dataset to provide new insights into gene function across a range of biological processes[16-21].

Here we leveraged insight from co-essentiality data to characterize an essential role for THAP1 in proteasome function. THAP1 is a ubiquitously expressed transcription factor which achieves sequence-specific DNA binding via an atypical THAP-type zinc finger domain located at its N-terminus[22]. Its target genes remain poorly defined, but THAP1 is thought to play an important role in DNA repair[23], cell cycle progression[24] and oligodendrocyte myelination[25,26]. Homozygous deletion of THAP1 leads to embryonic lethality[27,28]. Notably, a wide variety of autosomal dominant mutations located throughout the THAP1 coding sequence cause an early-onset form of the neurological disorder dystonia (DYT-THAP1, previously known as DYT-6), where progressive loss of motor function leads to sustained involuntary muscle contractions and abnormal posturing[29,30]. However, as the critical targets of THAP1 are poorly characterized, it remains unclear how the THAP1 mutations observed in dystonia patients result in disease.

Exploiting a fluorescent reporter knocked into the endogenous PSMB5 locus, here we demonstrate that the co-essential relationship between THAP1 and PSMB5 is explained by an essential role for THAP1 in activating PSMB5 expression. THAP1 binds to cognate sites within the PSMB5 promoter and is required for its basal expression, and hence loss of THAP1 results in insufficient PSMB5 expression, proteasome dysfunction and cell death. Finally, we leveraged our functional reporter assay to perform a deep mutational scan of THAP1, quantifying the activity of thousands of single amino acid variants to define the landscape of THAP1 mutations in dystonia.

## Results

### The dystonia gene THAP1 exhibits a co-essential relationship with the proteasome subunits PSMB5 and PSMB6

Leveraging co-essentially data from the DepMap project[15], we set out to characterize novel roles for genes involved in the UPS. Focusing on a manually curated set of ~1000 genes implicated in UPS function, we examined co-essential gene relationships derived from genome-wide CRISPR-Cas9 screens across ~1100 different cancer cell lines. Supporting the utility of this approach to identify genetic relationships that are functionally relevant, many of the most significant positive co-essential relationships clustered genes whose products are known to act in multi-protein complexes to facilitate protein degradation (Supplementary Fig. 1A). For instance, the RNF126 E3 ubiquitin ligase cooperates with BAG6 to target hydrophobic sequences mislocalised to the cytosol for proteasomal degradation[31] (Fig. 1A); Cul2, ElonginB/C and the von Hippel-Lindau (VHL) substrate adaptor comprise a Cullin-RING E3 ubiquitin ligase complex responsible for the degradation of hypoxia-inducible factor (HIF)-1α in normoxia (Fig. 1B)[32], and the CTLH complex is a multi-subunit E3 ligase orthologous to the yeast GID complex which degrades gluconeogenic enzymes[33] (Fig. 1C). Furthermore, several of the most significant negative co-essential relationships define E3 ligase-substrate pairs: for example, MDM2 mediates the

degradation of p53[34] (Supplementary Fig. 1B) and the Cul4 substrate adaptor AMBRA1 targets cyclin D[35-37] (Supplementary Fig. 1C).

Our follow-up work focused on the most statistically significant uncharacterized co-essential relationship in the dataset: THAP1 exhibits a highly significant positive association with both PSMB5 and PSMB6 (Fig. 1D–F). THAP1 is a transcription factor which binds DNA in a sequence-specific manner using a THAP-type zinc-finger domain, while PSMB5 and PSMB6 encode catalytic subunits of the proteasome core particle. Thus, we set out to test the hypothesis that the co-essential relationship between THAP1 and PSMB5/6 could be explained by an essential role for THAP1 in regulating the expression of catalytic proteasome subunits.

### Loss of THAP1 abrogates PSMB5 transcription

Lentiviral expression of Cas9 and CRISPR sgRNAs targeting THAP1 in HEK-293T cells was extremely toxic (Fig. 1G, H), consistent with Dep-Map data which demonstrates that knockout of THAP1 is broadly deleterious across cancer cell lines[15]. However, at day 5 post-transduction, before the onset of significant cell death, we found substantially reduced levels of PSMB5 transcripts by quantitative reverse transcription PCR (qRT-PCR) (Fig. 1I). In contrast, we observed no reduction in the expression of either PSMB6 or PSMB7, the other catalytic subunits of the proteasome (Fig. 1I). Concordantly, we also observed reduced abundance of PSMB5 protein as assessed by immunoblot (Fig. 1J). Thus, these data suggest that THAP1 is required to maintain basal levels of PSMB5 transcription.

### Lethality resulting from loss of THAP1 can be rescued by exogenous PSMB5

Next, we sought to test the hypothesis that the essentiality of THAP1 is due to its role in activating PSMB5 expression. Should this be the case, we reasoned that, irrespective of any reduction in the expression of endogenous PSMB5, an exogenous source of PSMB5 should rescue cell viability upon THAP1 ablation. Strikingly, unlike their wild-type counterparts, HEK-239T cells transduced with a lentiviral vector expressing PSMB5 did not display any significant growth defect following CRISPR/Cas9-mediated targeting of THAP1 (Fig. 2A and Supplementary Fig. 2A). Exogenous expression of PSMB6, however, was incapable of rescuing viability following THAP1 ablation (Fig. 2A and Supplementary Fig. 2B). Thus, the toxicity that results from loss of THAP1 is due to insufficient PSMB5 expression, explaining the molecular basis for their co-essential relationship. In contrast, we found no evidence to support a direct relationship between THAP1 and PSMB6, suggesting that their co-essential relationship arises indirectly through their shared relationship with PSMB5.

DepMap data demonstrates that disruption of THAP1 is broadly lethal across cancer cell lines (Fig. 2B). Thus, to generalize our finding that the essential requirement for THAP1 is to facilitate PSMB5 expression, we ablated THAP1 in three additional human cell lines: HeLa, A549 and THP-1. Mirroring our findings in HEK-293T cells, in A549 and HeLa we found that the toxicity observed upon loss of THAP1 could be ameliorated upon exogenous expression of PSMB5 (Fig. 2C, D). THP-1 cells, however, did not exhibit reduced viability following THAP1 disruption (Fig. 2E). This prompted us to examine in more detail the nature of the cell lines in which THAP1 is not essential, which were strikingly enriched ($P < 1 \times 10^{-9}$) for immune cells ('myeloid' or 'lymphoid' as defined by DepMap). Considering that the immunoproteasome is constitutively expressed by many immune cells[38,39], we reasoned that expression of PSMB8, the analogous counterpart of PSMB5 in the immunoproteasome, might relieve the essential requirement for THAP1. Indeed, there is a strong correlation between the essentiality of THAP1 and PSMB8 expression levels as measured by RNA-seq, wherein the cell lines in which THAP1 knockout has little or no impact on viability express the highest levels of PSMB8 (Fig. 2F). Indeed, we found that THP-1 cells expressed high levels of PSMB8 by

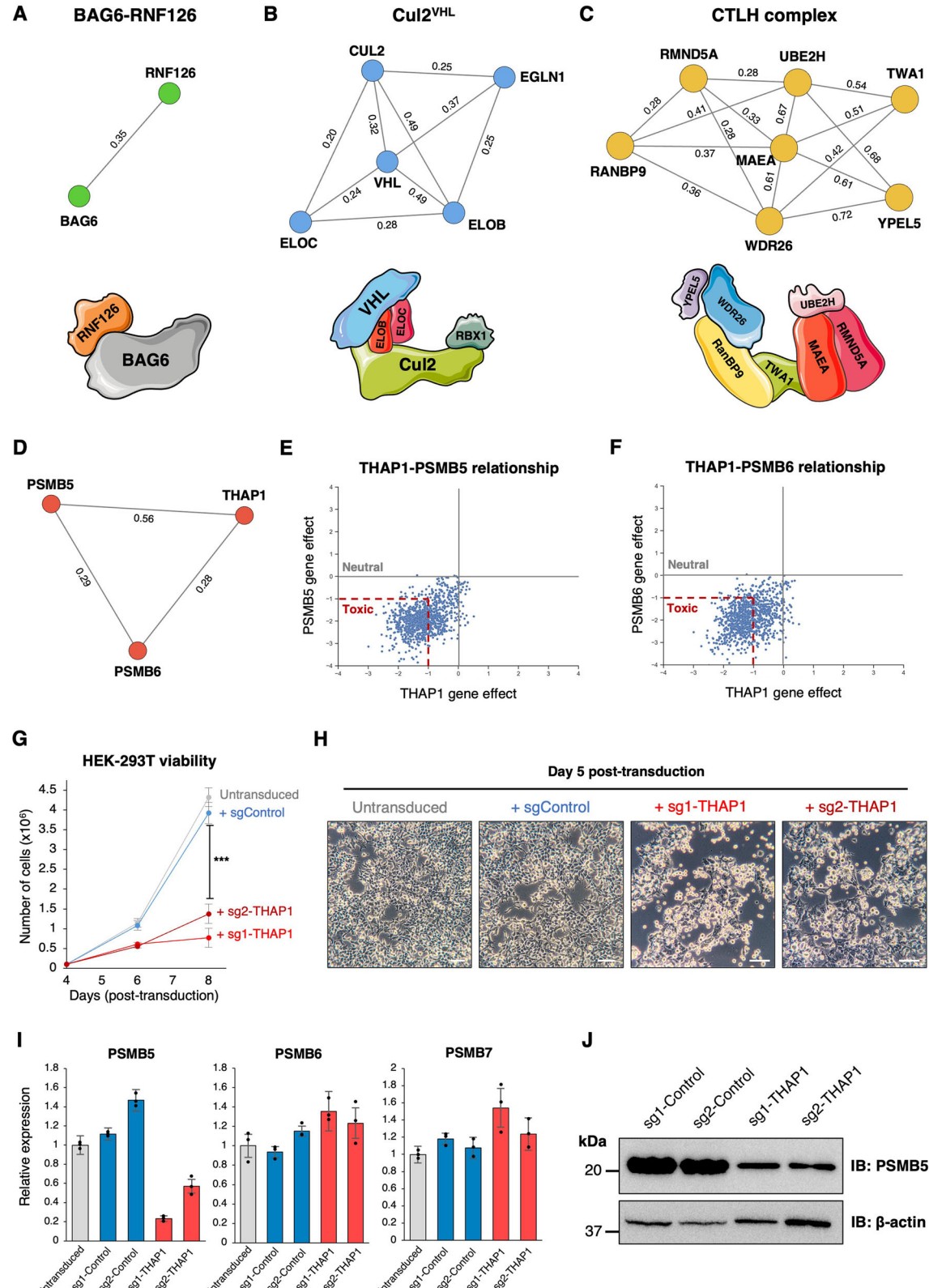

qRT-PCR (Fig. 2G). We further validated these conclusions in HEK-293T cells, where, like PSMB5, exogenous expression of PSMB8 maintained the viability of THAP1 knockout cells (Fig. 2H and Supplementary Fig. 2C). Thus, sustained expression of either PSMB5 or its immunoproteasome counterpart PSMB8 can rescue the toxicity associated with loss of THAP1.

## A fluorescent reporter at the endogenous *PSMB5* locus monitors THAP1 activity in live cells

We extended these findings by knocking-in a fluorescent reporter to the endogenous *PSMB5* locus, enabling us to monitor PSMB5 expression in live cells. Following transfection of HEK-293T cells with Cas9, an sgRNA targeting the transcriptional start site of *PSMB5* and a homology

**Fig. 1 | Transcriptional regulation of PSMB5 by THAP1 explains their co-essential relationship. A–C** Co-essential relationships involving UPS genes predict biological relationships, as exemplified by three E3 ligase complexes: the BAG6 complex (**A**), Cul2$^{VHL}$ (**B**) and the CTLH complex (**C**). Network diagrams were produced using NetworkX; numbers annotating the edges indicate pairwise correlation coefficients as calculated in ref. 16. **D–F** *THAP1* exhibits a strong positive co-essential relationship with both *PSMB5* and *PSMB6* across DepMap data. **G, H** *THAP1* disruption is toxic in HEK-293T cells. Cells were transduced with a lentiviral vector expressing Cas9

and the indicated sgRNAs, followed by puromycin selection to eliminate untransduced cells commencing 48 h later. A further 48 hours later, cells were counted, plated in equal numbers, and their viability assessed by counting (**G**) and brightfield microscopy (**H**). Data in (**G**) represent mean values of $n = 3$ biological replicates ± s.d. (\*\*\*$P < 0.001$, two-tailed t-test) (Scale bar = 100 μm). **I, J** Ablation of *THAP1* decreases PSMB5 expression. HEK-293T expressing Cas9 and the indicated sgRNAs were analyzed by qRT-PCR (**I**) and immunoblot (**J**). Data in (**I**) are presented as mean values of $n = 3$ technical replicates ± s.d. Source data are provided as a Source Data file.

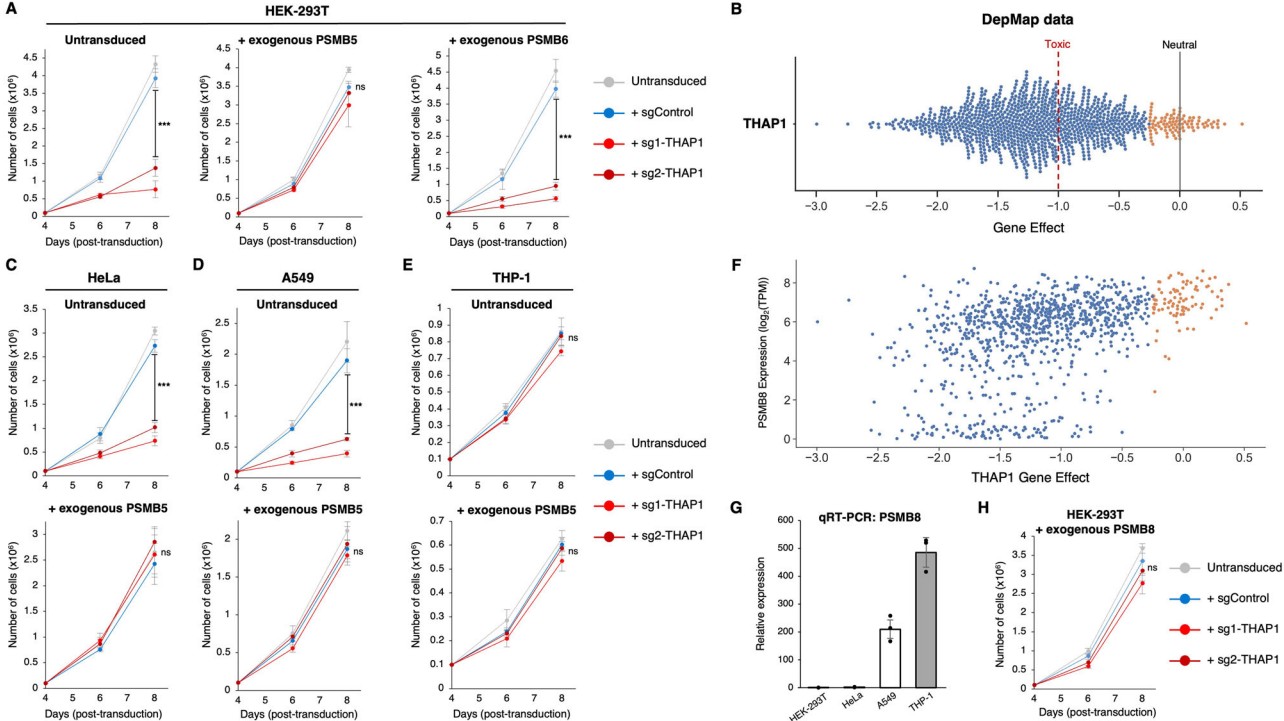

**Fig. 2 | Lethality resulting from THAP1 loss can be rescued by exogenous expression of PSMB5. A** Exogenous expression of PSMB5 rescues cell viability upon *THAP1* ablation. HEK-293T cells were first transduced with lentiviral vectors expressing either PSMB5 or PSMB6; then, following transduction with Cas9 and the indicated sgRNAs, cell numbers were monitored over time. **B** Loss of THAP1 is broadly toxic across cell types. CRISPR/Cas9-mediated ablation of *THAP1* adversely affects the viability of 946/1100 cancer cell lines (blue dots, representing effect scores < -0.25) examined by DepMap. **C, D** The toxicity associated with *THAP1* ablation is rescued by exogenous PSMB5 expression in HeLa cells (**C**) and A549 cells (**D**). **E–H** Like PSMB5, expression of PSMB8 also protects against the toxic effects of

THAP1 loss. THP-1 cells do not exhibit any substantial growth defect following *THAP1* ablation (**E**). High levels of PSMB8 expression are observed in the cell lines whose growth is not significantly affected by *THAP1* loss (orange dots, representing effect scores >-0.25) in DepMap data (**F**), and THP-1 cells strongly express PSMB8 as assessed by qRT-PCR (**G**). Exogenous expression of PSMB8 can rescue the viability of HEK-293T cells following *THAP1* disruption (**H**). Data in (**G**) represent mean values of $n = 3$ technical replicates ± s.d.; data in (**A**, **C–E**, and **H**) represent mean values of $n = 3$ biological replicates ± s.d. (\*\*\*$P < 0.001$, two-tailed t-test; ns, not significant). Source data are provided as a Source Data file.

donor vector encoding the green fluorescent protein (GFP) variant mClover3[40] followed by a P2A peptide (Fig. 3A), we were readily able to establish a population of cells (~10%) which were stably GFP-positive (Fig. 3B). Single cell clones isolated from the GFP-positive population (Fig. 3C) harbored GFP at the intended site as validated by PCR from genomic DNA (Supplementary Fig. 2D). Furthermore, lentiviral expression of shRNAs targeting PSMB5 resulted in a reduction in GFP expression (Supplementary Fig. 2E) prior to the onset of cell death (Supplementary Fig. 2F, G), validating that the reporter clones could be used to quantitatively assess PSMB5 expression.

We exploited our findings above to generate viable THAP1 knockout (KO) reporter cells. Following CRISPR/Cas9-mediated disruption of THAP1 in P$_{PSMB5}$-GFP reporter cells (sustained by exogenous expression of PSMB5) (Fig. 3D), we isolated single cell clones from the GFP$^{dim}$ population by fluorescence-activated cell sorting (FACS) (Fig. 3E). Disruption of the *THAP1* locus was confirmed by PCR from genomic DNA followed by Sanger sequencing (Supplementary Fig. 2H). Using primers specific to the 3' untranslated region of PSMB5 to allow

for selective detection of the endogenous transcript, we confirmed a substantial reduction in PSMB5 expression in two THAP1 KO clones by qRT-PCR (Fig. 3F). Our attempts to validate efficient knockout of THAP1 by immunoblot, however, were hampered by the paucity of effective commercial antibodies. In particular, we found that the Proteintech antibody (12584-1-AP) used in multiple previous studies detected a prominent band running around the expected molecular weight (~25 kDa), but whose abundance was not affected upon CRISPR/Cas9 targeting of *THAP1* (Fig. 3G). However, this antibody could readily detect exogenous THAP1 as a separate band that migrated just slightly slower, and, upon prolonged exposure, was able to detect endogenous THAP1 in control cells but not in the THAP1 knockout clones (Fig. 3G).

### THAP1 acts through cognate binding sites located within the *PSMB5* promoter

Next, we sought to determine how THAP1 might regulate PSMB5 expression. In support of a direct effect, chromatin immunoprecipitation followed by sequencing (ChIP-seq) data from the ENCODE

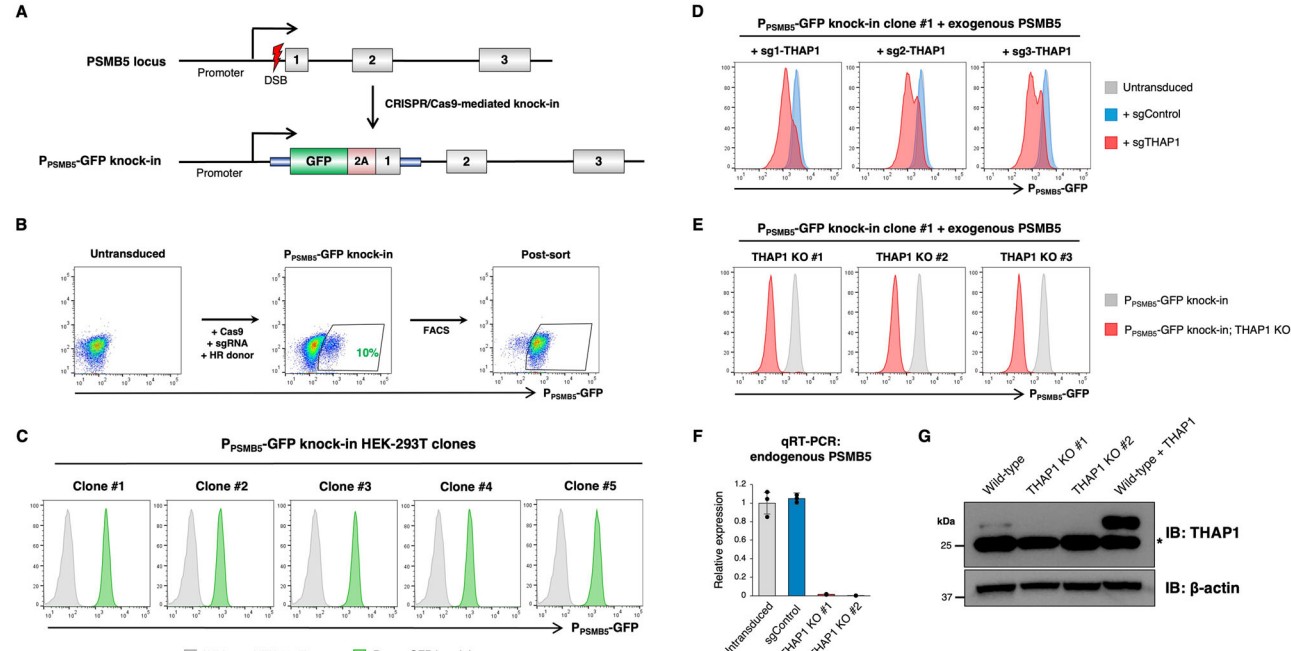

**Fig. 3 | A fluorescent reporter measures endogenous PSMB5 expression in live cells. A–C** CRISPR/Cas9-mediated knock-in of GFP into the endogenous PSMB5 locus. A schematic representation of the procedure is shown in (**A**). Transfection of HEK-293T cells with Cas9, an sgRNA targeting PSMB5 and a donor template resulted in ~10% GFP-positive cells (**B**), which were purified by FACS and single cell cloned (**C**). **D–G** THAP1 ablation reduces PSMB5 expression. CRISPR/Cas9-mediated disruption of *THAP1* in a $P_{PSMB5}$-GFP reporter clone (expressing exogenous PSMB5 to ensure viability) reduced $P_{PSMB5}$-GFP expression (**D**), permitting the derivation of GFP$^{dim}$ THAP1 KO clones (**E**). These THAP1 KO clones exhibited greatly reduced expression of PSMB5 by qRT-PCR (using primers annealing to the 3'UTR to ensure selective amplification of the endogenous transcripts) (**F**), and an absence of THAP1 protein by immunoblot (a non-specific band is indicated by an asterisk) (**G**). Data in (**F**) represent mean values of $n = 3$ technical replicates ± s.d. Source data are provided as a Source Data file.

project[41] revealed THAP1 occupancy immediately upstream of the *PSMB5* transcription start site (TSS) (Fig. 4A). Despite THAP1 binding to thousands of gene promoters (Supplementary Data 1), this property is not shared among the genes encoding proteasome subunits: *PSMD8* is the only other gene to exhibit THAP1 occupancy (Fig. 4A). THAP1 contains a THAP-type zinc finger domain which mediates sequence-specific DNA binding, and competitive EMSA experiments[42] have defined the consensus binding sequence ("THABS" motif) as TNNNGGCA (where N represents any nucleotide) (Fig. 4B). Strikingly, examination of the PSMB5 proximal promoter region revealed two perfect matches within 200 bp of the TSS, and a third near-perfect match a further ~500 bp upstream (Fig. 4C). Together, these data suggest that THAP1 binds cognate motifs in the PSMB5 promoter to activate its transcription.

We examined this hypothesis by engineering a lentiviral reporter system in which ~1 kb of the PSMB5 proximal promoter was placed upstream of GFP (Fig. 4D). Single copy expression of this reporter construct in HEK-293T cells resulted in robust GFP expression (Supplementary Fig. 3A). This appeared to be due in part to the activity of THAP1, as combined deletion of all three THABS motifs from the PSMB5 promoter ("ΔTHABS") resulted in decreased GFP expression (Fig. 4E). Importantly, CRISPR-mediated ablation of *THAP1* (performed following the introduction of exogenous PSMB5 to maintain cell viability) reduced GFP expression from the reporter construct driven by the wild-type PSMB5 promoter, but did not further reduce expression from the PSMB5 promoter lacking all THABS sites (Fig. 4F).

To investigate the relative importance of the three THABS motifs, we created an additional panel of mutant constructs in which each THABS motif was individually deleted. These data pointed to a critical role for site 2, as only the ΔTHABS2 construct exhibited decreased GFP expression relative to the level of the ΔTHABS construct (Supplementary Fig. 3B). Moreover, the ΔTHABS2 construct was the only one unaffected following ablation of THAP1, whereas a marked reduction

in GFP expression was observed in cells expressing ΔTHABS1 and ΔTHABS3 (Supplementary Fig. 3C). Thus, THAP1 binding to a cognate motif (THABS2) immediately upstream of the PSMB5 TSS appears critical for PSMB5 expression.

### Loss of THAP1 impairs proteasome function

As *PSMB5* encodes one of the three catalytic subunits of the constitutive 20S proteasome core particle, we set out to test the hypothesis that the toxicity associated with loss of THAP1 was due to proteasome dysfunction. First, we examined whether the catalytic activity of PSMB5 was required to sustain cell viability upon *THAP1* ablation. Whereas HEK-293T cells expressing wild-type PSMB5 did not exhibit any appreciable growth defect upon disruption of *THAP1*, cells expressing a catalytically-inactive PSMB5 mutant were not viable under these conditions (Fig. 5A). Second, as PSMB5 is critical to facilitate the integration of the other catalytic β subunits into the 20S core particle during proteasome assembly[43–45], we assessed whether loss of THAP1 resulted in defects in proteasome assembly. As a result of impaired autocatalytic cleavage of their N-terminal propeptide, an inability of the catalytic subunits to incorporate into the 20S core particle causes an accumulation of the immature proteins which can be detected by immunoblot[45]. Supporting the idea of defective proteasome assembly in the absence of THAP1, we found decreased abundance of the mature forms of PSMB6 and PSMB7, concomitant with the accumulation of immature (uncleaved) species (indicated by asterisks) which were absent from control cells (Fig. 5B). Moreover, native gels revealed the accumulation of proteasome assembly intermediates (indicated by asterisks) upon *THAP1* disruption which mirrored the defects observed upon PSMB5 knockdown (Fig. 5C). Finally, we reasoned that proteasome dysfunction upon THAP1 loss should result in the stabilization of short-lived proteins. Exploiting the Global Protein Stability (GPS) two-color lentiviral reporter system[46] (Fig. 5D), we found that CRISPR-mediated ablation of THAP1 stabilized two exogenous GFP-degron

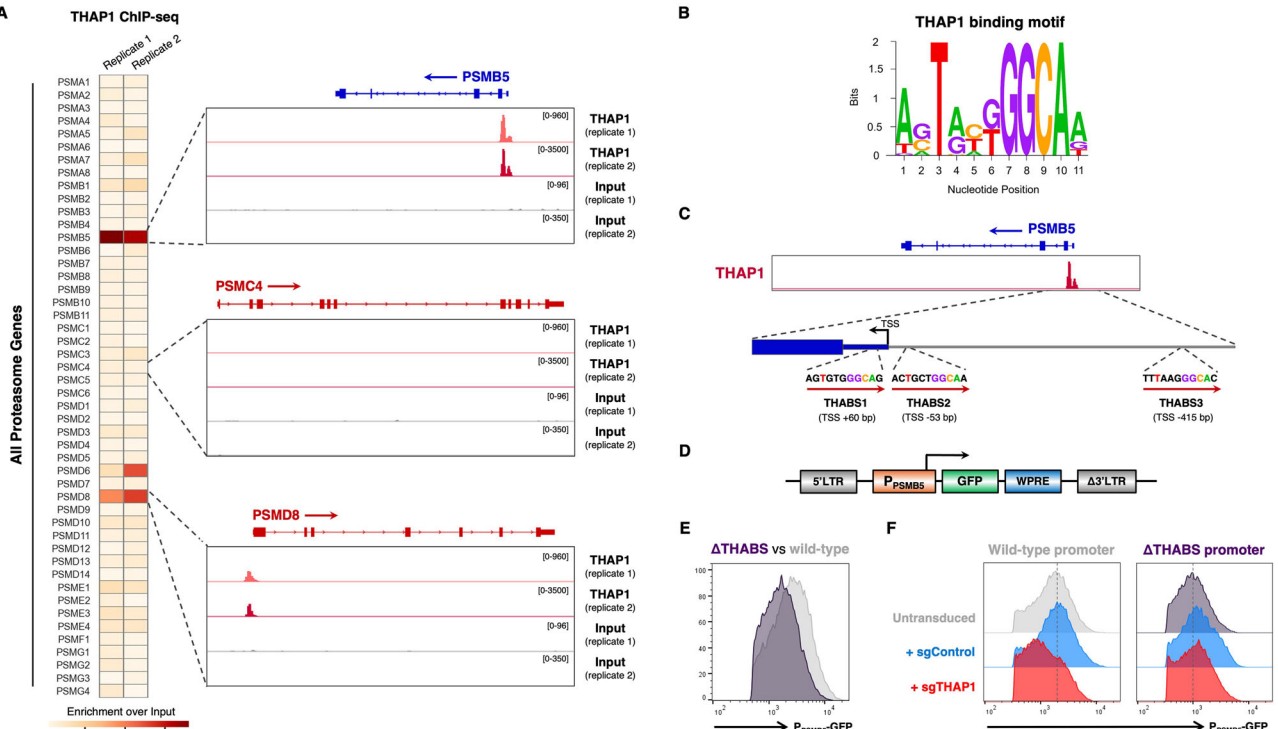

**Fig. 4 | THAP1 binds cognate motifs within the PSMB5 promoter to regulate its expression. A** THAP1 binds the *PSMB5* promoter. THAP1 ChIP-seq data in K562 cells[41] reveals an intense peak representing THAP1 occupancy at the *PSMB5* transcription start site (TSS) (top). PSMD8 is the only other proteasome subunit at which concordant binding of THAP1 is observed (bottom). **B** Consensus THAP1 binding site (THABS) motif (adapted from[68]). **C** Schematic representation of the three consensus THABS motifs located near the *PSMB5* TSS. **D**–**F** THAP1 targets cognate sites in the PSMB5 promoter to activate gene expression. **D** Schematic representation of a lentiviral reporter system in which -1 kb of the PSMB5 promoter drives the expression of GFP. Removal of the three THABS motifs ("ΔTHABS") reduced GFP expression (**E**); this effect was mediated through THAP1, as *THAP1* ablation decreased GFP expression driven by the wild-type promoter but not the ΔTHABS promoter (**F**). Source data are provided as a Source Data file.

fusion proteins to a similar degree as shRNA-mediated knockdown of PSMB5 (Fig. 5E). Similarly, *THAP1* disruption also resulted in increased abundance of endogenous HIF-1α, which is constitutively degraded by the proteasome in normoxia[32,47] (Fig. 5F). Altogether, these data support a model whereby the death of cells lacking THAP1 is caused by defective proteasome function resulting from inadequate expression of PSMB5.

**Defining the transcriptional targets of THAP1**

The key transcriptional targets of THAP1 remain poorly defined, hampering our ability to understand the functional consequences of THAP1 mutations in disease. Leveraging our insight that viable THAP1 knockout cells could be sustained by exogenous expression of PSMB5, we used RNA-seq to assess the impact of THAP1 deletion on the transcriptome. To avoid potential artefacts resulting from the analysis of single cell clones, we purified a population of THAP1 knockout cells. Following the CRISPR-mediated ablation of *THAP1* in P$_{PSMB5}$-GFP knock-in reporter cells (overexpressing PSMB5 to ensure viability), we isolated GFP$^{dim}$ cells by FACS and performed RNA-seq analysis (Fig. 6A). After discounting genes exhibiting differential expression between untransduced cells and cells expressing control sgRNAs, we identified 277 genes (220 downregulated, 57 upregulated) whose expression was significantly altered (FDR < 0.001, fold-change > 2) upon THAP1 knockout (Fig. 6B and Supplementary Data 2). Supporting the veracity of the dataset, the most significantly downregulated gene was Shieldin complex subunit 1 (*SHLD1*, previously known as *C20orf196*), consistent with recent findings describing a role for THAP1 in DNA double strand break repair choice[23]. Furthermore, although the requirement for exogenous expression of PSMB5 precluded its identification as a differentially expressed gene, the abundance of intronic reads mapping

to the endogenous PSMB5 locus was greatly reduced in the THAP1 knockout cells (Supplementary Fig. 4A-B).

To identify direct transcriptional targets of THAP1, we cross-referenced the differentially expressed genes identified through RNA-seq with THAP1 binding sites as defined by ChIP-seq. Among the differentially expressed genes, 42 exhibited THAP1 occupancy in their proximal promoter (Fig. 6C); of these 42 direct targets, 19 were downregulated upon loss of THAP1 while 23 were upregulated, suggesting that THAP1 has the potential to act as either a repressor or activator of transcription depending on the genomic context. However, the primary role of THAP1 appeared to be as an activator, with several of its direct targets exhibiting marked downregulation in its absence (Fig. 6B, C). We found no significant functional enrichment among the differentially expressed genes through GO term analysis, but their promoter sequences were enriched for transcription factor binding motifs for both THAP1 and YY1, a known THAP1 co-factor[23,48,49] (Fig. 6D and Supplementary Fig. 4C). None of these genes are currently associated with dystonia, but they include *ECH1*, an enzyme involved in fatty acid metabolism that has been previously identified as a THAP1 target[49], and *METTL3*, the N[6]-methyladenosine methyltransferase, which is an attractive therapeutic target in cancer[50]. Across five of the THAP1 target genes that exhibited the greatest degree of downregulation upon loss of THAP1, we further validated these findings by qRT-PCR (Fig. 6E–I). Altogether, these data define the genes directly targeted by the transcription factor activity of THAP1.

**A deep mutagenic scan defines the landscape of THAP1 mutations in Dystonia**

A wide range of autosomal dominant mutations distributed throughout the THAP1 coding sequence give rise to DYT-THAP1 dystonia[51–57],

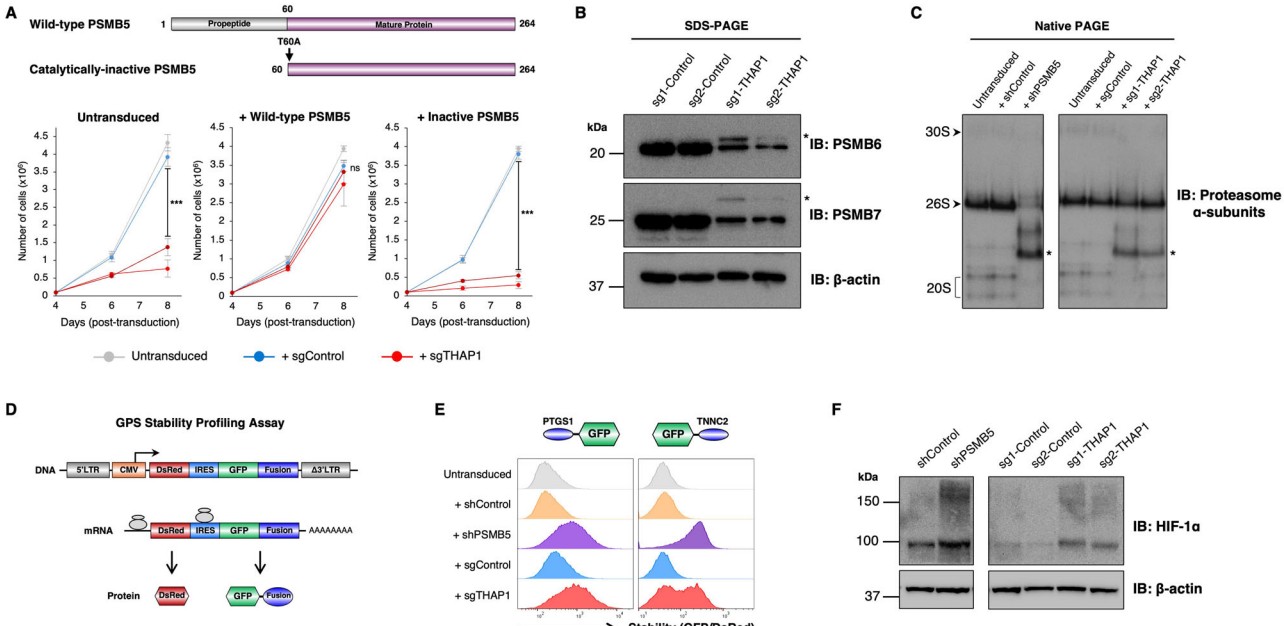

**Fig. 5 | PSMB5 insufficiency resulting from THAP1 loss impairs proteasome function. A** The catalytic activity of PSMB5 is required to rescue viability upon THAP1 loss. Exogenous expression of wild-type PSMB5, but not a catalytically-inactive mutant, restored the viability of HEK-293T cells following *THAP1* ablation. Data are presented as mean values of n=3 biological replicates +/− s.d. (***P < 0.001, two-tailed t-test; ns, not significant). **B, C** Loss of THAP1 impairs proteasome assembly. **B** *THAP1* ablation decreases the abundance of mature, processed PSMB6 and PSMB7 as assessed by immunoblot, but leads to the accumulation of the uncleaved proproteins (indicated with asterisks). **C** Native PAGE analysis reveals the accumulation of proteasome assembly intermediates (indicted with asterisks) following *THAP1* disruption, mimicking the defects observed upon PSMB5

knockdown. **D–F** THAP1 impairs the proteasomal degradation of short-lived proteins. **D** Schematic representation of the lentiviral Global Protein Stability (GPS) two-color fluorescent reporter system to monitor protein stability. **E** Stabilization of two model GFP-degron fusion proteins upon ablation of *THAP1*, as assessed by flow cytometry; the N-terminal peptide derived from PTGS1 harbors an N-terminal degron targeted by UBR-family E3 ligases[69], while the C-terminal peptide derived from TNNC2 harbors a C-terminal degron targeted by Cul4[DCAF12][70]. **F** Increased abundance of endogenous HIF-1α upon THAP1 disruption, as assayed by immunoblot. All immunoblot data is representative of at least two independent experiments. Source data are provided as a Source Data file.

an early-onset neurological disorder characterized by involuntary muscle contractions and movements causing abnormal and painful posturing. Thus, we sought to exploit our $P_{PSMB5}$-GFP knock-in reporter clone to assess the functional impact of DYT-THAP1 mutations. Genetic complementation of THAP1 KO cells with wild-type THAP1 did result in a restoration in $P_{PSMB5}$-GFP expression, although this effect was partial and did not restore GFP fluorescence to the levels observed in the parental cells (Supplementary Fig. 5A). However, this assay was sufficiently sensitive to report on THAP1 activity, as expression of an inactive THAP1 mutant unable to bind DNA (C5A, which abrogates zinc chelation by the zinc finger motif[22]) did not restore $P_{PSMB5}$-GFP expression (Supplementary Fig. 5A).

With the goal of globally defining how mutations in THAP1 affect its function, we leveraged our phenotypic assay in $P_{PSMB5}$-GFP knock-in reporter cells to carry out a deep mutational scan. Through microarray oligonucleotide synthesis, we generated a library of mutant constructs in which each residue of THAP1 (with the exception of the initiator methionine, 212 amino acids in total) was systematically replaced with all of the other 20 possible amino acids (Fig. 7A, B). The resulting site-saturation mutagenesis library was packaged into lentiviral particles and introduced into THAP1 KO $P_{PSMB5}$-GFP reporter cells at low multiplicity of infection, ensuring single-copy expression. We then used FACS to partition the population into GFP$^{dim}$ cells, in which no restoration of $P_{PSMB5}$-GFP expression was observed, and GFP$^{bright}$ cells, in which $P_{PSMB5}$-GFP expression was restored, and quantified the THAP1 variants present in each population by Illumina sequencing (Fig. 7B).

After an initial filtering step to remove variants with low read counts, we recovered data for 4002 of the 4240 possible single amino acid variants (94.3%). Overall, we observed high concordance

between mutant performance across two replicate experiments (Supplementary Fig. 5B); however, we discarded 179 mutants which exhibited discordant behavior between the two replicate experiments, leaving a total of 3823 (90.2%) variants for analysis (Supplementary Data 3). The results are summarized as a heatmap in Fig. 7C, with the data normalized such that the mean performance of all the control (wild-type) constructs is centered at 1; thus, the darker the red color the more deleterious the impact of the mutation on THAP1 function, whereas blue cells indicate mutations which may enhance the THAP1-mediated activation of $P_{PSMB5}$-GFP expression.

We evaluated the quality of the dataset in several ways. First, we considered residues essential for zinc chelation and hence folding of the zinc finger motif[22]: C5, C10, C54 and H57. These critical residues were uniformly essential for THAP1 activity, as mutation to any other residue prevented activation of the $P_{PSMB5}$-GFP reporter (Fig. 8A). Moreover, mutations across all residues previously determined to be important for DNA binding through biochemical assays[22] were extremely deleterious (Fig. 8B). Second, the global landscape of THAP1 activity correlated well with the predicted structure of THAP1 (Fig. 8C, D). In particular, mutation of most residues in the two structured regions, the THAP-type zinc finger (residues 2-81) and the predicted coiled-coil domain (residues 139–191) abrogated transcriptional activity, whereas most mutations targeting the unstructured central linker (residues 82–138) and C-terminus (residues 192–213) did not impair transcriptional activity (Figs. 7C and 8E). A notable exception, however, was the DHNY motif (residues 134–137) lying at the end of the central unstructured linker, which was absolutely critical for THAP1 function (Fig. 7C). Interestingly, this motif has been identified as the binding site for HCFC1[58], and AlphaFold 3[59] predicts with high confidence an interaction between the THAP1

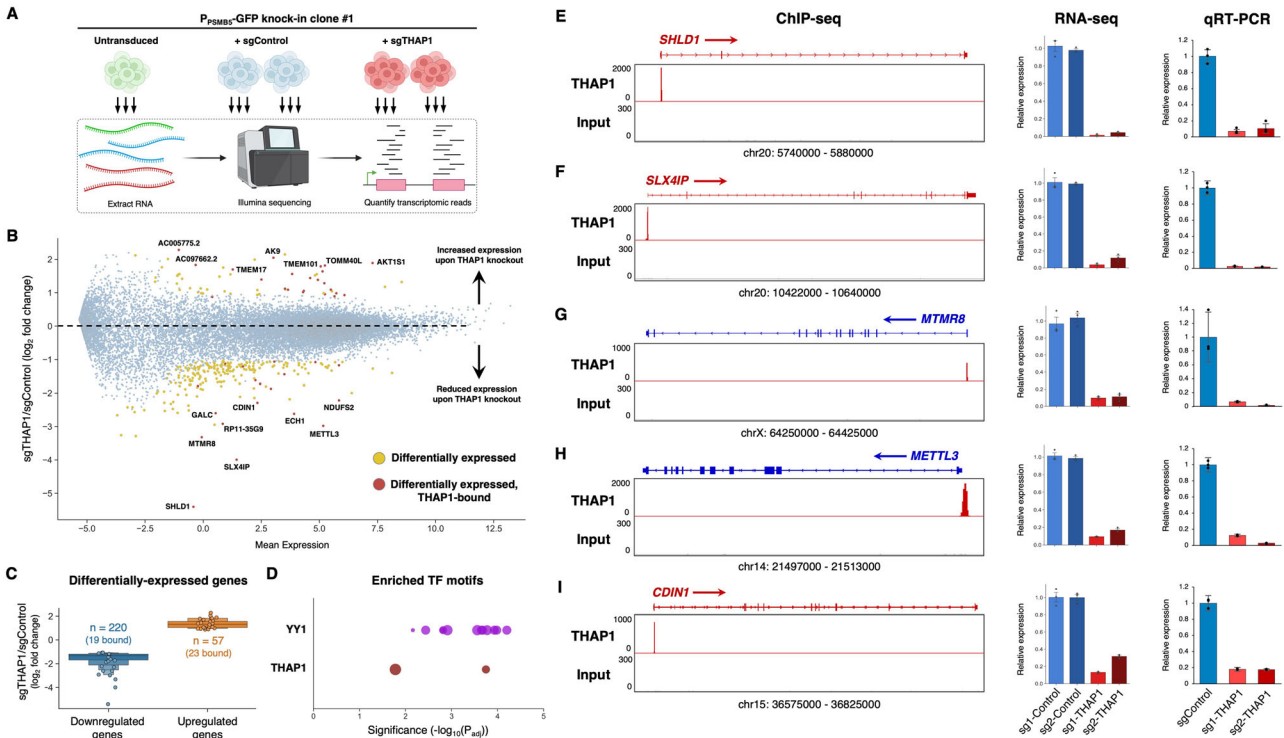

**Fig. 6 | Defining the transcriptional targets of THAP1. A** Schematic representation of the RNA-seq experiment. Created in BioRender. Timms, R. (2025) https://BioRender.com/ x85w665. **B, C** Identifying the transcriptional targets of THAP1. A summary of the RNA-seq dataset is shown in (**B**): genes exhibiting differential expression between sgControl and sgTHAP1 cells (n = 277) are highlighted in yellow, with the subset of those genes that display THAP1 occupancy as assessed by ChIP-seq (n = 42) colored red. Expression changes amongst all differentially expressed genes are summarized in (**C**), with circles representing genes bound by THAP1. **D** Consensus binding sites for YY1 and THAP1 are enriched amongst the promoters of the 42 direct THAP1 target genes. Each circle represents an individual YY1 (purple) or THAP1 (red) binding site (see also Supplementary Fig. 4C); bubble size is proportional to the number of the gene promoters containing the motif. **E–I** Validation of THAP1 target genes. Five target genes directly activated by THAP1 binding are shown: ChIP-seq data indicating THAP1 occupancy is shown on the left; the RNA-seq expression data is summarized in the center, and qRT-PCR validation is shown on the right. Data are presented as mean values of n = 3 technical replicates +/− s.d. Source data are provided as a Source Data file.

DHNY motif and the kelch repeats of HCFC1 (Supplementary Fig. 5C). HCFC1 has been identified as an essential cofactor for the THAP1-mediated activation of *SHLD*[23], and thus is also likely to be critical for the THAP1-mediated activation of *PSMB5*. Indeed, we confirmed that deletion of the DHNY motif and the coiled–coil domain, but not the disordered C-terminus, abolished the ability of THAP1 to activate the $P_{PSMB5}$-GFP reporter (Fig. 8F).

Many of the THAP1 mutations identified in dystonia patients remain of uncertain significance[57]. Thus, we examined the utility of this dataset in classifying the functional effects of THAP1 variants identified clinically (Fig. 8G). The majority of these mutations strongly impaired the ability of THAP1 to activate expression of the $P_{PSMB5}$-GFP reporter, consistent with the prevailing view that disease-causing mutations represent loss-of-function alleles[26,48,57]. However, some mutants exhibited activity at or approaching the level of the wild-type protein, suggesting that they might represent benign variants. To verify that the screen results could be faithfully recapitulated in individual experiments, we selected eight patient mutations predicted to abolish THAP1 activity (A7D, R13H, K24E, P26R, H57N, L72R, F81L and N136S) and compared their performance to five mutants predicted not to affect THAP1 activity (I80V, C83R, M143V, A166T and D192N). Validating the screen results, the eight inactive mutants exhibited little or no ability to activate $P_{PSMB5}$-GFP reporter expression (Fig. 8H and Supplementary Fig. 6), whereas the five active mutants exhibited similar performance to wild-type THAP1 (Fig. 8H). Altogether, these data illustrate structure-function relationships for THAP1 at high resolution, enabling the functional classification of clinical THAP1 mutations.

## Discussion

Co-essential relationships identified through the DepMap project[15] represent a rich resource to characterize gene function. Here we explain the co-essential relationship between THAP1 and PSMB5 by demonstrating that THAP1 is essential for the basal expression of PSMB5. Insufficient PSMB5 expression resulting from loss of THAP1 results in proteasome dysfunction and cell death, which can be rescued through exogenous expression of PSMB5. We exploit this finding to generate viable THAP1 knockout cells and hence identify transcriptional targets of THAP1 by RNA-seq. Finally, leveraging a phenotypic assay to systematically assess the activity of THAP1 mutants at the endogenous *PSMB5* locus, we define the transcriptional activity of THAP1 mutants found in dystonia patients. Overall, these data identify THAP1 as a regulator of proteasome function and suggest that aberrant proteostasis could be a factor underlying the pathogenesis of THAP1 dystonias.

Here we characterize THAP1 as an additional regulator of proteasome gene expression. In contrast to the master regulators Rpn4 and Nrf1[8–10], THAP1 appears to regulate only PSMB5. Why might THAP1 have evolved to exclusively regulate the expression of one single proteasome subunit? We speculate that perhaps there is a physiological circumstance wherein the downregulation of PSMB5 expression is beneficial, which could be achieved through the conditional inactivation of THAP1 activity. For example, it is plausible that transcriptional downregulation of *PSMB5* concomitant with upregulation of *PSMB8* might be beneficial upon viral infection, when immunoproteasomes are favored to increase the production of antigenic peptides[60,61], or during thymic development when PSMB11 (β5t) is incorporated in preference to PSMB5 and PSMB8 into the

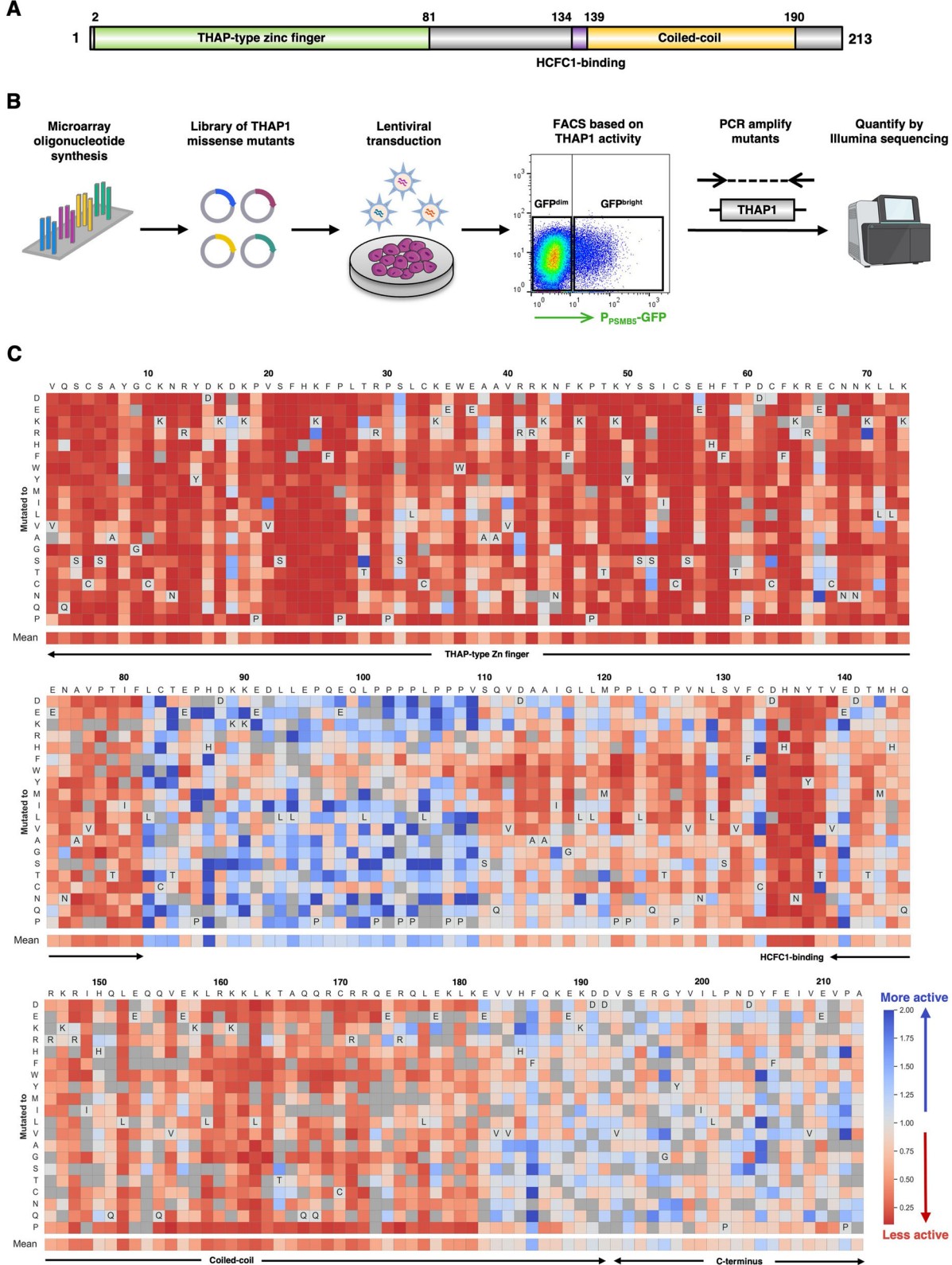

**Fig. 7 | A deep mutational scan defines the functional landscape of THAP1 mutations. A** Schematic representation of the domain architecture of the THAP1 protein. **B** Schematic representation of the deep mutational scan, designed to interrogate the ability of all possible single amino acid variants of THAP1 to activate expression of the endogenous $P_{PSMB5}$-GFP knock-in allele. Created in BioRender. Timms, R. (2025) https://BioRender.com/ x85w665. **C** Site-saturation mutagenesis reveals critical residues for THAP1 function. Each cell represents the performance of a single THAP1 mutant: the mean performance of all the wild-type proteins is centered at 1 (light gray), with red colors indicating mutants which abrogate activation of the $P_{PSMB5}$-GFP reporter and blue colors indicating mutants which may enhance the activation of the $P_{PSMB5}$-GFP reporter. Dark gray cells indicate mutants for which insufficient data was available for analysis.

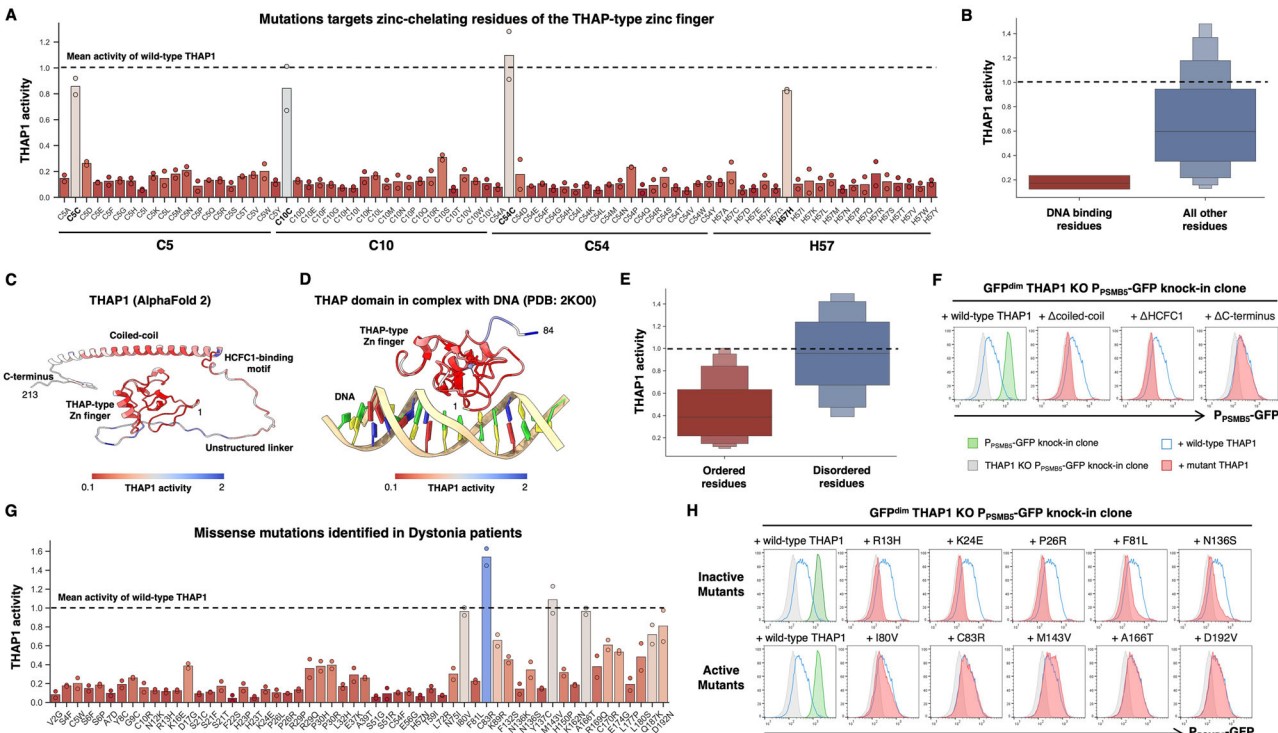

**Fig. 8 | Determining the functional effects of THAP1 mutations found in Dystonia patients. A, B** The deep mutational scan successfully identifies THAP1 residues critical for DNA binding. **A** Performance of all THAP1 variants targeting zinc-chelating residues of the zinc finger domain. Bars indicate the mean of two replicate experiments. Constructs which encode the wild-type protein are indicated in bold; the mean activity exhibited across all the wild-type THAP1 constructs is set at 1 (dotted line). **B** Distribution of activity scores across all the THAP1 variants targeting residues previously determined[22] to be important for DNA binding by THAP1. **C–F** Defining structure-function relationships for THAP1. Mean activity scores from the genetic screen were mapped onto the predicted structure of THAP1 (**C**) or the experimental structure of the THAP1 zinc finger domain[22] (**D**). Overall,

residues predicted to lie in ordered regions of the protein (AlphaFold pLDDT > 60) were much less tolerant of mutations than residues predicted to lie in disordered regions (**E**). Individual validation of the screen results was performed using flow cytometry: deletion of the coiled-coil and HCFC1-binding motif abrogated THAP1 function, whereas deletion of the disordered C-terminus did not (**F**). **G, H** Profiling the activity of THAP1 mutations found in Dystonia patients. **G** Performance of all missense variants identified in Dystonia patients, displayed as in (**A**). **H** Individual validation experiments measuring the activity of THAP1 mutants predicted to be inactive (top row) and THAP1 mutants predicted to be active (bottom row) by flow cytometry. See also Supplementary Fig. 5C–E.

thymoproteasome[62]. Another interesting question for future studies will be to examine whether the expression of other individual proteasome subunits (such as PSMB6 and PSMB7) is also subject to specific regulatory mechanisms.

The mechanisms through which THAP1 mediates its effects on gene expression remain unclear. THAP1 does not possess an obvious activator or repressor domain, and so it is likely that it acts through the recruitment of co-factors to target genes. A complex of THAP1 with YY1 and HCFC1 has previously been shown to mediate activation at the *SHLD* promoter[23], and the F81L dystonia mutation is thought to disrupt YY1 binding and hence impair THAP1-mediated transcriptional activation[49]. Our data offer support to this notion: YY1 binding motifs were strongly enriched amongst the direct targets of THAP1 identified by RNA-seq, and an intact HCFC1-binding motif was critical for THAP1-mediated activation of the P$_{PSMB5}$-GFP reporter. However, ChIP-seq reveals thousands of THAP1 binding sites in promoters across the genome, many of which colocalize with HCFC1 and YY1[23], and yet we observed relatively few transcriptional changes by RNA-seq. Thus, we speculate that other factors must be involved in determining whether THAP1 binding alters transcriptional activity. Our P$_{PSMB5}$-GFP knock-in reporter cells may therefore serve as a useful resource for further genetic interrogation of this pathway; for example, genome-wide CRISPR screens may identify additional genes required for the THAP1-mediated activation of *PSMB5*.

We leveraged our genetic reporter to characterize the impact of single amino acid variants on the ability of THAP1 to activate the

expression of *PSMB5*. This dataset, covering over 90% of all possible single amino acid variants, represents a rich resource for functional classification of THAP1 mutations. Specifically, these data strongly support the notion that disease-causing mutations in THAP1 generate loss-of-function alleles which are unable to regulate target gene expression: 84% of the missense mutations in THAP1 identified in dystonia patients exhibited performance at <50% of the wild-type protein. Thus, we propose that the mutations which do not impair THAP1 activity are likely to represent benign variants. Indeed, the clinical evidence supporting a pathogenic role for some THAP1 mutations remains equivocal[57]; for example, a previous study concluded that the I80V mutation was very likely to be benign, supported by the conservative nature of the substitution and the lack of any functional defect in a reporter assay[57].

How does loss of THAP1 function result in Dystonia? As the most plausible explanation is that dysregulated expression of one or more of its target genes leads to disease[63], our data advance progress towards answering this question in two ways. First, the identification of THAP1 as a critical activator of PSMB5 expression suggests that proteasome dysfunction could underlie the pathogenesis of DYT-THAP1, although, to the best of our knowledge, no other mutations in proteasome genes have so far been associated with dystonia. Second, by exploiting exogenous PSMB5 expression to generate viable THAP1 knockout cells, we were able to rigorously identify additional direct transcriptional targets of THAP1. However, as none of these genes are currently associated with dystonia and aberrant proteostasis is a

feature of many neurological disorders[64,65], these data highlight proteasome dysfunction as a candidate pathogenic mechanism underlying THAP1 dystonias.

# Methods

## Cell culture

HEK-293T, HeLa and A549 cells were grown in Dulbecco's Modified Eagle's Medium (DMEM, Merck #D6429); Jurkat and THP-1 cells were grown in Roswell Park Memorial Institute Medium (RPMI, Merck #R8758). Both were supplemented with 10% fetal bovine serum (ThermoFisher Scientific, #A5256701) plus penicillin and streptomycin (ThermoFisher Scientific, #15140122) and incubated at 37 °C plus 5% $CO_2$. All cells were routinely checked for mycoplasma contamination. Cell counting experiments were performed using a Countess II instrument (ThermoFisher Scientific) and analyzed using unpaired two tailed t-tests.

## Antibodies

Primary antibodies used in this study were: rabbit anti-THAP1 (Proteintech, #12584-1-AP, 1:5000), mouse anti-V5 tag (Abcam, #AB27671, 1:10000), rabbit anti-PSMB5 (Enzo Life Sciences, #BML-PW8895), mouse anti-PSMB6 (Enzo Life Sciences, #BML-PW8140, 1:5000), mouse anti-PSMB7 (Enzo Life Sciences, #BML-PW8145, 1:5000), mouse anti-HIF-1α (BD, #610959, 1:1000), mouse anti-proteasome α-subunits (Abcam, #22674, 1:1000), mouse anti-Vinculin (Sigma, #V9131, 1:10000) and mouse anti-β-actin (Sigma, #A2228, 1:10000). HRP-conjugated donkey anti-mouse IgG and donkey anti-rabbit IgG secondary antibodies were obtained from Jackson ImmunoResearch.

## Plasmids

Proteasome 20S core particle β-subunits were exogenously expressed from the pHRSIN-$P_{SFFV}$-GFP-WPRE-$P_{PGK}$-Blast$^R$/Hygro$^R$ lentiviral vectors (a gift from Paul Lehner), with constructs cloned in place of GFP via the Gibson assembly method using the NEBuilder HiFi Cloning Kit (NEB, #E5520S). GPS lentiviral vectors encoding the N-terminus of PTGS1 and the C-terminus of TNNC2 fused to GFP were gifts from Stephen Elledge. CRISPR sgRNA sequences were selected from the Brunello genome-wide library[66] and synthesized as top and bottom strand oligonucleotides (IDT). Oligos were phosphorylated (T4 PNK; NEB #M0201), annealed by heating to 95 °C followed by slow cooling to room temperature, and then inserted (T4 ligase; NEB #M0202) into lentiCRISPRv2 (Addgene #52961). shRNAs were cloned in an analogous manner into the pHR-SIREN-$P_{U6}$-shRNA-WPRE-$P_{PGK}$-Puro lentiviral vector (a gift from Paul Lehner) using the BamHI and EcoRI sites. Top strand oligonucleotide sequences used were:

sg1-Control (targets FOXP1 intron): caccgTGGGAACAGGATGAGGAAGG

sg2-Control (targets ATP1A1 intron): caccGATGGGCAAGAAGGAAGCAG

sg1-THAP1: caccgCTGCAAGAACCGCTACGACA

sg2-THAP1: caccGAAAACTGAGAGATTAACAG

sg3-THAP1: caccgCTGTGACCACAACTATACTG

shControl:gattcGTTATAGGCTCGCAAAAGGTTCAAGAGACCTTTTGCGAGCCTATAACTTTTTTg

shPSMB5:gattcCAATGTCGAATCTATGAGCTTCTCGAGAAGCTCATAGATTCGACATTGTTTTTTg

## Lentivirus production

Lentiviral stocks were generated through the transfection of HEK-293T cells with the specific lentiviral vector plus a mix of packaging plasmids encoding Gag-Pol, Rev, Tat and VSV-G. HEK-293T cells seeded at 70-90% confluence were transfected using PolyJet *In Vitro DNA* Transfection Reagent (SignaGen Laboratories, #SL100688) according to the manufacturer protocol. The media was replaced 24 h post-transfection and the viral supernatant was collected at 48 h post-transfection, centrifuged at $800 \times g$ for 5 min to remove cellular debris, and either applied immediately to target cells or stored at −80 °C in single-use aliquots.

## CRISPR/Cas9-mediated gene knock-in

A four-fragment Gibson assembly reaction was used to generate the homology donor vector. 5′ and 3′ homology arms (~1 kb) were amplified from genomic DNA, and were assembled together with a fragment encoding mClover3 followed by a P2A peptide and a pUC plasmid digested with PciI (NEB, #R0655) and SbfI (NEB, #R3642). The resulting plasmid was transfected into HEK-239T cells along with a PX459 (Addgene #48139, kindly deposited by Feng Zhang) plasmid encoding Cas9 and an sgRNA (CTTTCTGCCCACACTAGACA) targeting the start of the PSMB5 coding sequence. Transfected cells were selected with puromycin for 48 h commencing 24 h post-transfection. Two weeks later, cells that remained GFP$^+$ were single cell cloned by FACS.

## Flow cytometry and FACS

Analysis of cells by flow cytometry was performed using either an LSR-II or Fortessa instrument (BD Biosciences), collecting a minimum of 10,000 cells per sample. All flow cytometry data were collected through FACSDiva software and subsequently analyzed using FlowJo. Cell sorting was carried out using an Influx instrument (BD Biosciences).

## Immunoblotting

Cells were lysed in 1% SDS plus 1:200 Benzonase (Merck, #E1014) for 20 min at room temperature. Following the addition of Laemmli buffer (Bio-Rad, #161-0747), lysates were heated to 70 °C for 10 min. Proteins were separated by SDS-PAGE using 4-12% Bis-Tris gels (Merck, #MP41G12) and transferred onto an activated PVDF membrane (Merck, #IPFL00010). Membranes were blocked for a minimum of 30 min in 5% Skim Milk Powder (Merck, #70166) in PBS + 0.1% Tween-20 (PBS-T) (Merck, #P1379). Membranes were incubated with primary antibodies overnight at 4 °C, washed at least three times in PBS-T, and then incubated with HRP-conjugated secondary antibodies for 40 min at room temperature. Following a further five washes in PBS-T, reactive bands were visualized using SuperSignal West Detection Reagents (ThermoFisher Scientific, #32106, #34580 and #34076) and images collected on a ChemiDoc Imaging System (Bio-Rad). Raw images were processed using GNU Image Manipulation Platform (GIMP) version 2.10.34. Uncropped and unprocessed blot images are available in the Source Data file.

## Native PAGE

Cells were lysed in OK Lysis Buffer[67] containing 5% Digitonin (ThermoFisher Scientific, #BN2006) on ice for 20 min, and, following addition of Tris-Glycine Native Sample Buffer (ThermoFisher Scientific, #LC2673), samples were separated using NuPAGE Tris-Acetate 3-8% gels (ThermoFisher Scientific, #EA03752PK2) with Tris-Glycine Native Running Buffer (ThermoFisher Scientific, #LC2672). Protein denaturation was achieved by soaking the gel in solubilization buffer[67] for 15 min; the subsequent transfer, blocking and immunoblotting steps were performed as described above.

## Imaging

HEK-293T cells were imaged on a Zeiss Primovert Inverted Phase Contrast Microscope Ph1/0.3 at 10x magnification using the NexYZ 3-axis Universal Smartphone Adapter (Celestron).

## qRT-PCR

Total RNA was extracted from ~1 million cells using the RNeasy Mini Kit (Qiagen, #74104) with QIAshdredder Mini Spin Columns (Qiagen, #79656) as per the manufacturer's protocol, including on-column DNaseI digestion using the RNase-Free DNase Set (Qiagen, #79254).

Reverse transcription was performed with 1 μg of RNA using one-step reaction using LunaScript RT SuperMix Kit (NEB, #E3010) as indicated by the manufacturer. For subsequent analysis by qPCR, 1 μl of cDNA template, 0.5 μl of each primer (10 μM) and 12.5 μl Luna Universal Probe qPCR Master Mix (NEB, M3004) were mixed in a final volume of 25 μl; thermocycling was performed on a QuantStudio 7 Flex Real-Time PCR system (ThermoFisher Scientific). Relative expression was quantified using the ΔΔCt method relative to RPS18; data are expressed as mean ± standard deviation and $P$ values calculated using a one-tailed unpaired t-test. All qPCR amplicons were verified using agarose gel electrophoresis. Primer sequences are listed in Supplementary Data 4.

### RNA-seq

RNA extracted as above was sent to Azenta for strand-specific polyA+ Illumina library preparation and sequencing. Raw sequence reads were trimmed of adaptor sequence using Cutadapt (version 4.1), aligned using HISAT2 (version 2.2.1) to the human genome (GRCh38 genome_tran index), and further analyzed using SeqMonk (version 1.48.1).

### Deep mutational scan

The THAP1 coding sequence was divided into six segments for mutagenesis (encompassing residues 2–37, 38–73, 74–109, 110–145, 146–181 and 182–213). Using pKLV2-$P_{EF1\alpha}$-THAP1-$P_{PGK}$-Puro-T2A-BFP-WPRE as the starting point, six vectors were generated in which 'stuffer' regions flanked by BbsI restriction sites replaced the sequence encoding each segment. An oligonucleotide pool encoding the mutant alleles was synthesized by Twist Bioscience: for each segment, each amino acid was systematically exchanged to all 20 possible amino acids. Each of the six mutant segments were amplified from the oligonucleotide pool by PCR (Q5, NEB #M0491L), gel purified (QIAEX II Gel Extraction Kit, Qiagen #20021), and then cloned into their respective 'stuffer' vector cut with BbsI (NEB, #R3539S) using the Gibson assembly method (NEB, #E5520S). The reaction products were electroporated into DH10β cells (ThermoFisher Scientific, #18290015) and grown on LB plates with ampicillin overnight at 30 °C; the next morning, plasmid DNA was extracted (GenElute HP Plasmid Midiprep Kit, Merck #NA0200-1KT) from all of the E. coli and verified by Sanger sequencing (Azenta).

The six mutant pools were combined into three for screening (1–2, 3–4, and 5–6). These were packaged into lentiviral particles and, in duplicate, introduced into GFP^dim THAP1 KO $P_{PSMB5}$-GFP reporter cells at a multiplicity of infection of ~0.3 (~30% BFP+ cells). Five days post-transduction, the BFP+ cells were partitioned into GFP^dim (THAP1 inactive) and GFP^bright (THAP1 active) populations by FACS. Genomic DNA was extracted from the sorted cells (Gentra Puregene Cell Kit, Qiagen #158767) and the exogenous THAP1 sequences in each sample amplified by PCR (Q5, NEB #M0493L), using primers annealing to invariant regions flanking each mutagenized segment. PCR products were purified using a spin column (QIAquick PCR Purification Kit, Qiagen #28104), and then used as a template for a second PCR reaction using primers to add the Illumina adaptors and indexes. Products were purified using a spin column, quantified using a Nanodrop spectrophotometer, and mixed evenly; the final pool was purified from a 2% agarose gel (QIAEX II Gel Extraction Kit, Qiagen #20021). All steps were performed at sufficient scale to maintain at least 200-fold representation of the library. Sequencing was performed on an Illumina NovaSeq 6000 instrument using 150 bp paired-end reads. Count tables quantifying the abundance of each mutant in each sorting bin were generated by trimming the raw sequence reads of constant flanking sequence using Cutadapt (version 4.1) and aligning them to a reference index using Bowtie 2 (version 2.4.5).

### Reporting summary

Further information on research design is available in the Nature Portfolio Reporting Summary linked to this article.

## Data availability

THAP1 RNA-seq data has been deposited in NCBI's Gene Expression Omnibus with the GEO Series accession number GSE264536 and Illumina sequencing data from the THAP1 deep mutagenic scan are available in NCBI's Sequence Read Archive (SRA) with the accession number PRJNA1102672. THAP1 ChIP-seq data was obtained from NCBI's GEO with the accession number GSM803408. DepMap datasets are publicly available at https://depmap.org/portal/download/all/. Source data are provided with this paper.

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

## Acknowledgements

We are grateful to Gabriela Grondys-Kotarba and Reiner Schulte at the Cambridge Institute for Medical Research Flow Cytometry Core Facility and to the NIHR Cambridge BRC Cell Phenotyping Hub. This work was supported by an Academy of Medical Sciences Springboard Grant (SBF007\100019) and an Isaac Newton Trust/Wellcome Trust ISSF/University of Cambridge Joint Research Grant to R.T.T. R.T.T. is a Pemberton-Trinity Fellow.

## Author contributions

D.E.R. and R.T.T. conceived the study, and, along with D.W.G., performed the experiments and analyzed the data. D.E.R. and R.T.T wrote the manuscript.

## Competing interests

The authors declare no competing interests.
