## [Transparent Peer Review file · Nature Communications]

Loss-of-function mutations in the dystonia gene THAP1 impair proteasome function by inhibiting PSMB5 expression

Corresponding Author: Dr Richard Timms

Version 0:

Reviewer comments:

Reviewer #1

(Remarks to the Author)

Proteasomal degradation is vital in all cells and organisms, and dysfunction or failure of proteasomal degradation is associated with various human diseases, including cancer and neurodegeneration. Here, the authors report from a previous dataset (DepMap) that the transcription factor THAP1 has a co-essential relationship with PSMB5 gene. They further confirm that THAP1 is essential for the basal expression of PSMB5, with THAP1-mediated loss of PSMB5 resulting in proteasome dysfunction and cell death. This defect can be rescued by exogenous expression of either PSMB5 or PSMB8, its equivalent subunit within the immunoproteasome. They next went on to define the transcriptional targets of THAP1 by RNA-seq. Finally, they performed a phenotypic assay to survey PSMB5 regulation by THAP1 mutants, highlighting the impact of disease-relevant mutations (Dystonia) on PSMB5 levels.

Data presented in the manuscript support overall authors' conclusion, although some specific conclusions may need to be strengthened (see below). On the whole, the paper is well written, timely and will bring significant advances to the field of proteasome and neurodegeneration. The main novelties lie on (1) the discovery that THAP1 selectively controls PSMB5 transcription (possibly PSMD8 as well) resulting in proteasome impairment, which may also be occurring in patients with Dystonia, and (2) that exogenous expression of PSMB5 restores proteostasis in THAP1-deficient cells, as this could be a therapeutic way forward in treatment of Dystonia (e.g., treatment with IFN-gamma). In summary, I do think that this paper will be a good addition to the scientific literature, and I would support publication of this manuscript providing the more specific concerns listed below are addressed.

Major concerns:

- Level of proteasome subunits doesn't always correlates with the level of assembled proteasome. It would be important to test the impact of THAP1 loss on proteasome assembly and activity using native gels. This will also strengthen data from Figure 5 suggesting that THAP1 loss results in proteasome assembly defects. While the accumulation of proteasome precursors is indicative of proteasome assembly defects, the visualisation of assembled proteasomes and their assembly intermediates (that are known to accumulate upon assembly defects) by native gels is a more direct way to evidence a defect in proteasome assembly.

- PSMB8 also rescues loss of THAP1, it would be interesting to know if treatment inducing immunoproteasome subunits such as IFN-gamma can also rescue the loss of THAP1, as this could bring perspectives on new potential therapeutic avenues.

- It doesn't seems like statistical analyses have been conducted for various figures, where even the number of biological replicates or standard deviation are not shown (figure) or mentioned (legend) (e.g., Figures 1G, 2A,C,H...). Therefore, statistical analyses should be performed to further support the authors' conclusion and be clearly stated in the figures and figure legends.

- Probe for PSMB5 (and GFP) in Fig. 3G to confirm protein levels are decreased in THAP1 KO clones

- It would be interesting to interrogate published dataset from patients with THAP1 mutation (Dystonia) and see if PSMB5

transcript is commonly downregulated compared to other proteasome genes.

Minor points:

- Introduction could expand a bit on what is known about the transcriptional regulation of proteasome genes, even if this is mentioned in the discussion.
- The function of the proteasome resides in the assembly of the 33 distinct subunits to form a function complex, the authors should discuss their view on why only two core subunits show co-essential relationship, as loss of any other proteasome subunits are expected to have a similar phenotype (reduced proteasome function)?
- Are all other proteasome genes not regulated by THAP1 without TNNGGCA motif? This would highlight whether it is just the presence of the motif or not that dictates THAP1-mediated regulation of proteasome genes, or other regulation mechanisms are involved.
- It would be interesting to probe for a general marker of proteasome degradation as well such as K48-linked ubiquitin chains (WT versus THAP1 KO).
- I would reformulate the sentence: "Armed with the knowledge that exogenous PSMB5 expression ameliorates the toxicity associated with THAP1 loss, we reasoned that we could leverage our findings to assess the impact of THAP1 deletion on the transcriptome.", as this is rather counterintuitive as exogenous PSMB5 expression restores protein degradation and will have more impact on protein levels (proteomics) rather than transcripts. Moreover, Fig. 6 is not really necessary to me, it disrupts the flow of the story and, while interesting, brings no new aspects on the role of THAP1 on regulating proteasome function which is the main point of the story. I would personally remove it but this is the authors' decision.
- Are any other proteasome gene mutations been reported in Dystonia? As other proteasome gene mutations are likely to trigger similar proteasome impairment. This should be discussed.

Reviewer #2

(Remarks to the Author)

This is a very good manuscript. Starting from a robust DepMap correlation between THAP1 and PSMB5, the authors show that THAP1 transcription factor regulates the expression of PSMB5, an essential subunit of the proteasome. Strikingly, exogenous expression of PSMB5 can rescue the lethality of THAP1 knockout cells. To my knowledge, there are very few examples of human transcription factors with such a specific, genetically dissectable target (ZNF410 that activates the NuRD subunit CHD4 comes to mind; maybe worth mentioning this and possibly others in the discussion). The authors further characterize the effects of THAP1 knockout on gene expression and proteasome function, and also conduct a deep mutational scanning of THAP1. The DMS results make sense in the light of the known structural features of THAP1 and known disease mutations in the protein.

The results are relevant for our understanding of the regulation of proteasome assembly and for understanding the pathogenic mechanisms behind DYT-THAP1 dystonia.

Overall, the experiments are well conducted, the results are very clear, and the conclusions are fully supported by the data. Honestly, I have a hard time coming up with further improvements. I only have a few things that I suggest the authors address before publication.

- 1) Many mutations in the DMS experiment seem to increase the activity of THAP1, in particular in the linker. Could the authors comment on the potential mechanisms of action of these?
- 2) I don't quite understand how the TF motif enrichment is calculated in Figure 6D. Shouldn't there be just one value for the enrichment of THAP1 and YY1 binding sites across the target genes?
- 3) AlphaFold3 seems to predict an interface between the THAP1 DHNY motif and HCFC1. Maybe the authors want to take a look at this and include it in the manuscript if the interface looks good?
- 4) It would be nice to add more discussion about the relative roles of PSMB5 vs the shieldin complex underlying DYT-THAP1 dystonia.

Reviewer #3

(Remarks to the Author)

In this work, Ramage et al. investigated the effects of loss of THAP1 on the ubiquitin-proteasome system (UPS) using HEK293, HeLa, A549 and THP-1 cells. Using coessentiality data from the DepMap project, they described an essential role of the dystonia gene THAP1 in

maintaining basal expression of PSMB5.

The authors propose a model according to which the death of cells lacking THAP1 is caused by defective proteasome function due to insufficient expression of PSMB5.

The main criticism is that the authors used almost exclusively HEK293 cells to investigate the link between the loss of THAP1 and UPS and to further investigate the effects of DYT-THAP1 causing mutations on PSMB5 without validating their data in cells derived from DYT-THAP1 patients.

Less importantly, HeLa cells lack PARKIN, a gene encoding a ubiquitin E3 ligase that is involved in neurodegenerative disorders such as Parkinson's disease through UPS-mediated mitophagy.

In the results section, the entire first paragraph and Figures 1A-C could be deleted. It is unclear what relevance this has to the work presented.

In Figure 1, the authors should quantify the knockdown efficiency of THAP1 in HEK293 cells transduced with sgTHAP1_1 and sgTHAP1_2 by qPCR. Their HEK293-CRISPR cultures are heterogeneous cultures consisting of cells expressing various forms of THAP1, including inframe InDels, stable frameshift protein forms, etc., which could contribute to cell toxicity. Frameshift-induced nonsense-mediated mRNA decay could occur in only a subset of cells (and most likely affect only one allele). Alternatively, the authors should use the shRNA approach here.

It is particularly interesting that PSMB8 can rescue the viability of cells caused by the loss of THAP1 and the reduction of PSMB5. However, while PSMB8 mRNA levels in HEK293 and HeLa cells are below the limit of detection, PSMB8 levels in A549 cells are only 50% lower compared to THP-1 cells, while the toxicity caused by THAP1 loss is comparable between HEK293, HeLa and A549 cells. Can you explain this discrepancy?

To prove that THAP1 toxicity acts only by reducing PSMB5 levels in HEK293, HeLa and A549, the authors should replicate cell toxicity by knocking down PSMB5 in HEK293, HeLa and A549 and by knocking down PSMB8 in THP-1 cells lacking THAP1. In Figure S2E, where the authors use shRNA against PSMB5, no toxicity was reported when validating their knockin GFP-2A-PSMB5 reporter cells from HEK293.

The authors should also specify which 2A peptide they used, as they may differ in their self-cleavage ability.

In Figure S2F, three alleles can be seen in the PPSMB5-GFP knock-in THAP1 KO clone #1, which should not be possible. Furthermore, it is impossible to accurately identify edited alleles in heterozygous cells or mixed cell populations using Sanger sequencing. The authors should check the knockout efficiency by qPCR as well. It is not uncommon that InDels in ex1 do not initiate NMD. Please use standard nomenclature to describe the mutations in your purified clones.

It is a pity that the authors studied the effects of THAP1 mutations on PSMB5 using HEK293 cells (expressing PSMB5), even though they could accurately distinguish the expression of endogenous and exogenous PSMB5. In my opinion, using THP-1 GFP-A2-PSMB5 would be a more elegant solution to study the THAP1-PSMB5 axis.

Is there any other way to create a catalytically inactive PSMB5 variant, e.g. by introducing one or more point mutations. I suspect that removing the first 60aa of a protein has more serious consequences (misincorporation, degradation...) for the protein than just losing its activity.

Defining the transcriptional targets of THAP1 is a nice asset to the paper but does not help in further strengthening link between THAP1- PSMB5. In terms of role of THAP1 in dystonia it would be more appropriate to use neural cells such as neural progenitor cells or neuroblastoma (eg. SH-SY5Y) cells. In comparison to HEK293 neural cells express set of genes whose differential expression between control and THAP1 KO or DYT-THAP1 mutation could be more relevant for neurological disorder such dystonia..

The authors reported that overexpressed wildtype THAP1 could not restore normal GFP expression in their PPSMB5-GFP knock-in reporter clone. This could suggest that adverse effect of their THAP1 manipulation on PPSMB5 is not exclusively due to loss of THAP1 but due to gain of function of truncated/mutated THAP1 variant caused by CRISPR/Cas9. Therefore it is essential to demonstrate loss of mRNA by qPCR in their model.

Defining the transcriptional targets of THAP1 is a nice addition to the work, but does not help to further strengthen the link between THAP1 and PSMB5. Regarding the role of THAP1 in dystonia, it would be more appropriate to use neural cells such as neural progenitor cells or neuroblastoma cells (e.g. SH-SY5Y). Compared to HEK293, neural cells express a number of genes whose differential expression between control and THAP1 KO or DYT-THAP1 mutation cells may be more relevant to neurological disorders such as dystonia.

The authors reported that overexpressed wild-type THAP1 failed to restore normal GFP expression in their PPSMB5-GFP knock-in reporter clone. This may indicate that the adverse effects of their THAP1 manipulation on PPSMB5 are not solely due to the loss of THAP1, but to the gain of function of the truncated/mutated THAP1 variant caused by CRISPR/Cas9. Therefore, it is important to detect the loss of mRNA by qPCR in their model.

Reviewer comments:

Reviewer #1

(Remarks to the Author)

I thank the authors for their detailed response to questions. I think that the authors address my concerns in a satisfactory manner, and that their data now better support their conclusions. The statistical issue has been corrected. To this end, I have no further concerns and support publication.

Reviewer #2

(Remarks to the Author)

The authors have addressed all my relevant points and the manuscript is now ready for publication in my opinion. I'd like to congratulate the authors on an elegant study!

Reviewer #1

Proteasomal degradation is vital in all cells and organisms, and dysfunction or failure of proteasomal degradation is associated with various human diseases, including cancer and neurodegeneration. Here, the authors report from a previous dataset (DepMap) that the transcription factor THAP1 has a co-essential relationship with PSMB5 gene. They further confirm that THAP1 is essential for the basal expression of PSMB5, with THAP1-mediated loss of PSMB5 resulting in proteasome dysfunction and cell death. This defect can be rescued by exogenous expression of either PSMB5 or PSMB8, its equivalent subunit within the immunoproteasome. They next went on to define the transcriptional targets of THAP1 by RNA-seq. Finally, they performed a phenotypic assay to survey PSMB5 regulation by THAP1 mutants, highlighting the impact of disease-relevant mutations (Dystonia) on PSMB5 levels.

Data presented in the manuscript support overall authors' conclusion, although some specific conclusions may need to be strengthened (see below). On the whole, the paper is well written, timely and will bring significant advances to the field of proteasome and neurodegeneration. The main novelties lie on (1) the discovery that THAP1 selectively controls PSMB5 transcription (possibly PSMD8 as well) resulting in proteasome impairment, which may also be occurring in patients with Dystonia, and (2) that exogenous expression of PSMB5 restores proteostasis in THAP1-deficient cells, as this could be a therapeutic way forward in treatment of Dystonia (e.g. treatment with IFN-gamma). In summary, I do think that this paper will be a good addition to the scientific literature, and I would support publication of this manuscript providing the more specific concerns listed below are addressed.

We would like to thank the reviewer for their positive feedback on the manuscript.

Major concerns:

- Level of proteasome subunits doesn't always correlates with the level of assembled proteasome. It would be important to test the impact of THAP1 loss on proteasome assembly and activity using native gels. This will also strengthen data from Figure 5 suggesting that THAP1 loss results in proteasome assembly defects. While the accumulation of proteasome precursors is indicative of proteasome assembly defects, the visualisation of assembled proteasomes and their assembly intermediates (that are known to accumulate upon assembly defects) by native gels is a more direct way to evidence a defect in proteasome assembly.

We are very grateful to the reviewer for this helpful suggestion. We have now performed native PAGE analysis following THAP1 disruption, which reveals the accumulation of proteasome assembly intermediates mimicking those observed upon knockdown of PSMB5. We believe that these data significantly strengthen the case that loss of THAP1 results in

proteasome assembly defects resulting from PSMB5 insufficiency, and so we have added a new Figure 5C in the revised manuscript:

Figure 5 | PSMB5 insufficiency resulting from THAP1 loss impairs proteasome function. (C) Loss of THAP1 impairs proteasome assembly. Native PAGE analysis reveals the accumulation of proteasome assembly intermediates (indicated with asterisks) following *THAP1* disruption, mimicking the defects observed upon PSMB5 knockdown.

The modified text reads:

‘Moreover, native gels revealed the accumulation of proteasome assembly intermediates (indicated by asterisks) upon *THAP1* disruption which mirrored the defects observed upon PSMB5 knockdown (Fig. 5C).’

- PSMB8 also rescues loss of THAP1, it would be interesting to know if treatment inducing immunoproteasome subunits such as IFN-gamma can also rescue the loss of THAP1, as this could bring perspectives on new potential therapeutic avenues.

This is an interesting suggestion, but unfortunately we have not been able to demonstrate rescue of the toxicity associated with loss of THAP1 upon IFN γ treatment (although interpretation of this experiment is complicated by the growth arrest induced by IFN γ). However, we agree that compounds that augment PSMB5/8 expression or activate proteasome function could be considered as potential therapeutic options.

- It doesn't seem like statistical analyses have been conducted for various figures, where even the number of biological replicates or standard deviation are not shown (figure) or mentioned (legend) (e.g., Figures 1G, 2A,C,H...). Therefore, statistical

analyses should be performed to further support the authors' conclusion and be clearly stated in the figures and figure legends.

We have now performed three independent biological replicates for all experiments monitoring cell viability over time (Figures 1G, 2A, 2C-E, 2H, 5A and Fig. S2F) and added error bars and statistical analysis to all of these graphs. As an example, the updated Fig. 2A now looks like this:

Figure 2 | Lethality resulting from THAP1 loss can be rescued by exogenous expression of PSMB5. (A) Exogenous expression of PSMB5 rescues cell viability upon *THAP1* ablation. HEK-293T cells were first transduced with lentiviral vectors expressing either PSMB5 or PSMB6; then, following transduction with Cas9 and the indicated sgRNAs, cell numbers were monitored over time. (***) $P < 0.001$, t-test; ns, not significant)

- Probe for PSMB5 (and GFP) in Fig. 3G to confirm protein levels are decreased in THAP1 KO clones

As the viability of the THAP1 KO clones is maintained by exogenous PSMB5 expression, immunoblot analysis for PSMB5 would be unable to distinguish between the endogenous and exogenous proteins. Instead, we performed qRT-PCR using primers binding to the 3'UTR (which was omitted from the exogenous construct) to selectively assess the abundance of the endogenous PSMB5 transcript, which confirmed that endogenous PSMB5 expression was downregulated in response to THAP1 KO (Fig. 3F). For direct assessment of GFP levels in THAP1 KO clones, flow cytometry (Fig. 3E) is a more quantitative method than immunoblot analysis.

- It would be interesting to interrogate published dataset from patients with THAP1 mutation (Dystonia) and see if PSMB5 transcript is commonly downregulated compared to other proteasome genes.

This is an interesting suggestion, but to our knowledge there are no suitable RNA-seq datasets from DYT-THAP1 patients that would allow us to address this question. Interestingly, as far as we are aware PSMB5 has not been identified as a THAP1 target in any

of the RNA-seq analyses carried out in experimental systems; this may potentially reflect compensatory changes which the cells have undergone in order to maintain viability upon THAP1 loss.

Minor points:

- Introduction could expend a bit on what is known about the transcriptional regulation of proteasome genes, even if this is mentioned in the discussion.

We are grateful to the reviewer for this helpful suggestion; we have moved the paragraph on this topic that was previously in the Discussion upfront into the Introduction where we agree it would fit better.

- The function of the proteasome resides in the assembly of the 33 distinct subunits to form a function complex, the authors should discussed their view on why only two core subunits show co-essential relationship, as loss of any other proteasome subunits are expected to have a similar phenotype (reduced proteasome function)

Interestingly, proteasome subunits do not exhibit significant co-essential relationships with all other proteasome subunits in DepMap data. For example, PSMB5, PSMB6 and PSMB7 only share significant co-essential relationships with each other, but not with any other proteasome subunits. Thus, although loss of any proteasome subunit presumably results in impaired proteasome function, the growth phenotypes resulting from loss of different classes of proteasome subunits must be somewhat different. In light of this, is it not surprising that THAP1 only exhibits co-essential relationships with PSMB5 and PSMB6 but not any other proteasome subunits.

Are all others proteasome genes not regulated by THAP1 without TNNGGCA motif? This would highlight whether it is just the presence of the motif or not that dictates THAP1-mediated regulation of proteasomes genes, or other regulation mechanisms are involved.

We are certain that other regulatory mechanisms must be involved, as the presence of the TNNGGCA motif alone is not a good predictor of THAP1 transcriptional activity. This relatively small target sequence is found frequently throughout the genome; THAP1 binds thousands of these sites located in promoters as assessed by ChIP-seq, but we observe transcriptional changes at only a handful of the downstream genes following loss of THAP1 by RNA-seq. Thus we would hypothesise that colocalization of THAP1 with other transcription factors and/or co-factors at target gene promoters must be required for altered transcriptional activity.

- It would be interesting to probe for a general marker of proteasome degradation as well such as K48-linked ubiquitin chains (WT versus THAP1 KO).

We would like to thank the reviewer for this suggestion. We have assessed the overall abundance of ubiquitin by immunoblot using the VU-1 antibody, wherein THAP1 loss does appear to increase ubiquitin conjugates globally. However, as our positive control condition (PSMB5 knockdown) only itself resulted in modest accumulation of ubiquitin conjugates, we have concluded that this assay is not a sufficiently sensitive measure of proteasome function and hence we have decided not to include these data in the revised manuscript.

Reviewer Response Figure 1. Assessing the effect of THAP1 ablation on the accumulation of ubiquitin conjugates in HEK-293T cells through immunoblot analysis.

- I would reformulate the sentence: “Armed with the knowledge that exogenous PSMB5 expression ameliorates the toxicity associated with THAP1 loss, we reasoned that we could leverage our findings to assess the impact of THAP1 deletion on the transcriptome.”, as this is rather counterintuitive as exogenous PSMB5 expression restore protein degradation and will have more impact on protein levels (proteomics) rather than transcripts.

We apologise for the confusion here. We were trying to convey that an RNA-seq experiment comparing wild-type and THAP1 knockout cells would not normally be possible given the death observed upon THAP1 knockout, but would be enabled following exogenous PSMB5

expression. We have hopefully made this clearer by altering this sentence in the revised manuscript as follows:

‘Leveraging our insight that viable THAP1 knockout cells could be sustained by exogenous expression of PSMB5, we used RNA-seq to assess the impact of THAP1 deletion on the transcriptome.’

- Moreover, Fig. 6 is not really necessary to me, it disrupts the flow of the story and, while interesting, brings no new aspects on the role of THAP1 on regulating proteasome function which is the main point of the story. I would personally remove it but this is the authors’ decision.

We agree with the reviewer that the RNA-seq data in Fig. 6 is tangential to the main thrust of the paper, but we feel that these data are an important contribution to furthering understanding of the THAP1 target genes which are likely to be dysregulated in dystonia patients. Thus, although their inclusion does somewhat disrupt the flow of the primary story, we feel that it is important that they are included.

- Are any other proteasome gene mutations been reported in Dystonia? As other proteasome gene mutations are likely to trigger similar proteasome impairment. This should be discussed.

To the best of our knowledge, no other mutations in proteasome genes have been associated with Dystonia. The data presented in this manuscript implicate, for the first time, proteasome dysfunction in the pathogenesis DYT-THAP1.

In the Discussion of the revised manuscript we have expanded the following sentence accordingly:

‘First, the identification of THAP1 as a critical activator of PSMB5 expression suggests that proteasome dysfunction could underlie the pathogenesis of DYT-THAP1, although, to the best of our knowledge, no other mutations in proteasome genes have so far been associated with dystonia.’

Reviewer #2

This is a very good manuscript. Starting from a robust DepMap correlation between THAP1 and PSMB5, the authors show that THAP1 transcription factor regulates the expression of PSMB5, an essential subunit of the proteasome. Strikingly, exogenous expression of PSMB5 can rescue the lethality of THAP1 knockout cells. To my knowledge, there are very few examples of human transcription factors with such a specific, genetically dissectable target (ZNF410 that activates the NuRD subunit CHD4 comes to mind; maybe worth mentioning this and possibly others in the discussion). The authors further characterize the effects of THAP1 knockout on gene expression and proteasome function, and also conduct a deep mutational scanning of THAP1. The DMS results make sense in the light of the known structural features of THAP1 and known disease mutations in the protein.

The results are relevant for our understanding of the regulation of proteasome assembly and for understanding the pathogenic mechanisms behind DYT-THAP1 dystonia.

Overall, the experiments are well conducted, the results are very clear, and the conclusions are fully supported by the data. Honestly, I have hard time coming up with further improvements. I only have a few things that I suggest the authors address before publication.

We are very grateful to the reviewer for such positive comments.

1) Many mutations in the DMS experiment seem to increase the activity of THAP1, in particular in the linker. Could the authors comment on the potential mechanisms of action of these?

Perhaps the most plausible explanation would be that they may increase the stability (half-life) and hence abundance of the THAP1 protein (although increased affinity for DNA target sites and/or co-factors for example may also be possible). We would be reluctant to speculate on their potential mechanism of action, however, without having performed further experiments to test these hypotheses.

2) I don't quite understand how the TF motif enrichment is calculated in Figure 6D. Shouldn't there be just one value for the enrichment of THAP1 and YY1 binding sites across the target genes?

We apologise for the confusion here. The database used for this analysis considers multiple related motifs for each transcription factor individually (see the full output shown in Fig.

S4C), and hence the summary plot in Fig. 6D is depicting one circle per motif. We have clarified this in the revised manuscript by modifying the figure legend for Figure 6D:

Figure 6 | Defining the transcriptional targets of THAP1.

(D) Consensus binding sites for YY1 and THAP1 are enriched amongst the promoters of the 42 direct THAP1 target genes. Each circle represents an individual YY1 (purple) or THAP1 (red) binding site (see also Fig. S4C); bubble size is proportional to the number of the gene promoters containing the motif.

3) AlphaFold3 seems to predict an interface between the THAP1 DHNY motif and HCFC1. Maybe the authors want to take a look at this and include it in the manuscript if the interface looks good?

We would like to thank the reviewer for this helpful suggestion. We have included the following in the revised manuscript as the new Fig. S5C:

Supplementary Figure 5 | A deep mutagenic scan defines the functional consequences of THAP1 mutations.

(C) AlphaFold 3 prediction of the interaction between the THAP1 DHNY motif (gold) and residues (magenta) of the HCFC1 kelch repeats. Hydrogen bonds (cyan) predicted by ChimeraX are shown.

4) It would be nice to add more discussion about the relative roles of PSMB5 vs the shieldin complex underlying DYT-THAP1 dystonia.

We would be reluctant to speculate further on a possible role for the Shieldin complex in dystonia. Shinoda et al. (2021) were careful to point out in their 'Limitation of the Study' section that there was no evidence that unresolved DNA damage contributes to abnormal neuronal function in DYT-THAP1 dystonia, and as such we feel we should defer to these authors and refrain from further discussion on this point.

Reviewer #3

In this work, Ramage et al. investigated the effects of loss of THAP1 on the ubiquitin-proteasome system (UPS) using HEK293, HeLa, A549 and THP-1 cells.

Using coessentiality data from the DepMap project, they described an essential role of the dystonia gene THAP1 in maintaining basal expression of PSMB5.

The authors propose a model according to which the death of cells lacking THAP1 is caused by defective proteasome function due to insufficient expression of PSMB5.

The main criticism is that the authors used almost exclusively HEK293 cells to investigate the link between the loss of THAP1 and UPS and to further investigate the effects of DYT-THAP1 causing mutations on PSMB5 without validating their data in cells derived from DYT-THAP1 patients.

We accept the reviewer's criticism regarding the scope of the study and we concede that we have not validated our findings in the context of dystonia patients. However, the primary goal of the project was to further our understanding the fundamental mechanisms through which proteasome gene expression is regulated, and we believe our manuscript provides strong support for an important role for THAP1 in this process via PSMB5. Although many of the experiments were carried out in HEK-293T cells which can be easily manipulated, we have confirmed our key findings in multiple other cell lines and DepMap data illustrates the importance of the THAP1-PSMB5 relationship across over one thousand cell lines.

Less importantly, HeLa cells lack PARKIN, a gene encoding a ubiquitin E3 ligase that is involved in neurodegenerative disorders such as Parkinson's disease through UPS-mediated mitophagy.

We are confused as to the relevance of a potential role for Parkin in this context.

In the results section, the entire first paragraph and Figures 1A-C could be deleted. It is unclear what relevance this has to the work presented.

We appreciate the reviewer's opinion here, but we feel that Figure 1A-C are important to validate the utility of using DepMap data to identify functionally relevant relationships between UPS genes. Thus, we have decided to keep these sections in the revised manuscript.

In Figure 1, the authors should quantify the knockdown efficiency of THAP1 in HEK293 cells transduced with sgTHAP1_1 and sgTHAP1_2 by qPCR. Their HEK293-CRISPR cultures are heterogeneous cultures consisting of cells expressing various forms of THAP1, including inframe InDels, stable frameshift protein forms, etc., which could contribute to cell toxicity. Frameshift-induced nonsense-mediated mRNA decay could occur in only a subset of cells (and most likely affect only one allele). Alternatively, the authors should use the shRNA approach here.

qRT-PCR is not an appropriate method to quantify the efficiency of CRISPR/Cas9-mediated gene disruption, as indels introduced following repair of Cas9-induced cleavage events may completely abolish protein function without affecting mRNA abundance. Validation of THAP1 gene disruption in the knockout clones is presented in Fig. S2F by Sanger sequencing and Fig. 3G by immunoblot.

It is particularly interesting that PSMB8 can rescue the viability of cells caused by the loss of THAP1 and the reduction of PSMB5. However, while PSMB8 mRNA levels in HEK293 and HeLa cells are below the limit of detection, PSMB8 levels in A549 cells are only 50% lower compared to THP-1 cells, while the toxicity caused by THAP1 loss is comparable between HEK293, HeLa and A549 cells. Can you explain this discrepancy?

We agree that this is a surprising result. Our hypothesis would be that the level of PSMB8 expression in A549 cells still remains below a critical threshold needed to prevent the toxicity associated with loss of THAP1, but we have not investigated this effect further.

To prove that THAP1 toxicity acts only by reducing PSMB5 levels in HEK293, HeLa and A549, the authors should replicate cell toxicity by knocking down PSMB5 in HEK293, HeLa and A549 and by knocking down PSMB8 in THP-1 cells lacking THAP1.

CRISPR/Cas9-mediated ablation of THAP1 was consistently toxic in HEK-293T, HeLa and A549 cells, as described in Figures 1 and 2. Furthermore, exogenous expression of PSMB5 in each of these cell lines (Fig. 2A, C, D and E) ensured viability following THAP1 depletion, proving that PSMB5 expression alone is sufficient to rescue the deleterious effect of THAP1 loss. These data demonstrate that the toxicity resulting from loss of THAP1 is due to PSMB5 insufficiency.

In Figure S2E, where the authors use shRNA against PSMB5, no toxicity was reported when validating their knockin GFP-2A-PSMB5 reporter cells from HEK293.

We apologise for this omission and thank the reviewer for bringing this to our attention. We consistently find that shRNA-mediated knockdown of PSMB5 is extremely toxic. We have added data assessing the viability of HEK-293T cells following PSMB5 knockdown to the revised manuscript as Fig. S2F and S2G:

Supplementary Figure 2 | Validation of manipulated HEK-293T cell lines.

(F-G) PSMB5 knockdown is toxic in HEK-293T cells. Cells were transduced with a lentiviral vector expressing the indicated shRNAs. 48 hours post-transduction, cells were counted, plated in equal numbers, and their viability assessed by counting (F) and brightfield microscopy (G). (Scale bar = 100 μ m) (***) $P < 0.001$, t-test)

The modified text reads:

‘Furthermore, lentiviral expression of shRNAs targeting PSMB5 resulted in a reduction in GFP expression (**Fig. S2E**) prior to the onset of cell death (**Fig. S2F-G**), validating that the reporter clones could be used to quantitatively assess PSMB5 expression.’

The authors should also specify which 2A peptide they used, as they may differ in their self-cleavage ability.

We employed the P2A peptide, which has been shown to have the highest efficiency among the 2A peptides¹. This detail has been added to the revised manuscript:

‘Following transfection of HEK-293T cells with Cas9, an sgRNA targeting the transcriptional start site of *PSMB5* and a homology donor vector encoding the green fluorescent protein (GFP) variant mClover3 followed by a P2A peptide (**Fig. 3A**), we were readily able to establish a population of cells (~10%) which were stably GFP-positive (**Fig. 3B**).’

In Figure S2F, three alleles can be seen in the PPSMB5-GFP knock-in THAP1 KO clone #1, which should not be possible. Furthermore, it is impossible to accurately identify edited alleles in heterozygous cells or mixed cell populations using Sanger sequencing. The authors should check the knockout efficiency by qPCR as well. It is not uncommon

that InDels in ex1 do not initiate NMD. Please use standard nomenclature to describe the mutations in your purified clones.

HEK-293T cells are considered pseudotriploid³, and hence the three mutated THAP1 alleles that we observe is anticipated.

It is a pity that the authors studied the effects of THAP1 mutations on PSMB5 using HEK293 cells (expressing PSMB5), even though they could accurately distinguish the expression of endogenous and endogenous PSMB5. In my opinion, using THP-1 GFP-A2-PSMB5 would be a more elegant solution to study the THAP1-PSMB5 axis.

In hindsight, we agree with the reviewer that generating the GFP-P2A-PSMB5 knock-in in a cell line expressing PSMB8 would have been advantageous.

Is there any other way to create a catalytically inactive PSMB5 variant, e.g. by introducing one or more point mutations. I suspect that removing the first 60aa of a protein has more serious consequences (misincorporation, degradation...) for the protein than just losing its activity.

The catalytically-inactive PSMB5 mutant we used has a point mutation (T60A) which abolishes threonine protease activity. However, as the first 59 residues of PSMB5 constitute a pro-peptide which is removed by auto-cleavage prior to incorporation into the proteasome, the T60A mutation would prevent this processing event. Therefore, we consider the most minimal inactive PSMB5 mutant with the best chance of assembling into proteasomes to be the combined $\Delta 59$ -T60A mutant which we employed^{4,5}.

Defining the transcriptional targets of THAP1 is a nice asset to the paper but does not help in further strengthening link between THAP1- PSMB5. In terms of role of THAP1 in dystonia it would be more appropriate to use neural cells such as neural progenitor cells or neuroblastoma (eg.SH-SY5Y) cells. In comparison to HEK293 neural cells express set of genes whose differential expression between control and THAP1 KO or DYT-THAP1 mutation could be more relevant for neurological disorder such dystonia.

We agree with the reviewer that to better understand the transcriptional targets of THAP1 relevant to dystonia pathogenesis it would have been preferable to have carried out the RNA-seq experiment in a neuronal cell line. However, exploiting the P_{PSMB5}-GFP reporter to purify THAP1 knockout cells was critical to the success of our RNA-seq experiment in HEK-293T cells, and hence we feel that the establishment of a similar system in other cell lines would be required before the proposed experiment could be performed effectively.

The authors reported that overexpressed wildtype THAP1 could not restore normal GFP expression in their PPSMB5-GFP knock-in reporter clone. This could suggest that adverse effect of their THAP1 manipulation on PPSMB5 is not exclusively due to loss of

THAP1 but due to gain of function of truncated/mutated THAP1 variant caused by CRISPR/Cas9. Therefore it is essential to demonstrate loss of mRNA by qPCR in their model.

As discussed above, qRT-PCR is not an appropriate method to assess the efficiency of CRISPR/Cas9-mediated gene disruption. Although we do not understand why exogenous THAP1 does not fully restore P_{PSMB5}-GFP expression in THAP1 knockout clones to the level observed in the parental cells, we would consider this far more likely to be due to an inadequacy of the exogenously-expressed THAP1 rather than a defect related to the endogenous THAP1 which has been disrupted. Furthermore, we have fully characterised the edited THAP1 locus by Sanger sequencing as described in Fig. S2H.

Reviewer Response References

1. Szymczak, A. L. *et al.* Correction of multi-gene deficiency in vivo using a single ‘self-cleaving’ 2A peptide-based retroviral vector. *Nature Biotechnology* 2004 22:5 **22**, 589–594 (2004).
2. Bajar, B. T. *et al.* Improving brightness and photostability of green and red fluorescent proteins for live cell imaging and FRET reporting. *Scientific Reports* 2016 6:1 **6**, 1–12 (2016).
3. Lin, Y. C. *et al.* Genome dynamics of the human embryonic kidney 293 lineage in response to cell biology manipulations. *Nature Communications* 2014 5:1 **5**, 1–12 (2014).
4. Shi, C. X. *et al.* Proteasome subunits differentially control Myeloma cell viability and proteasome inhibitor sensitivity. *Mol Cancer Res* **18**, 1453 (2020).
5. Li, X., Kusmierczyk, A. R., Wong, P., Emili, A. & Hochstrasser, M. β -Subunit appendages promote 20S proteasome assembly by overcoming an Ump1-dependent checkpoint. *EMBO J* **26**, 2339 (2007).